# Thermocline state change in the Eastern Equatorial Pacific during the late Pliocene/early Pleistocene intensification of Northern Hemisphere Glaciation

Kim A. Jakob[1], Jörg Pross[1], Christian Scholz[1], Jens Fiebig[2], Oliver Friedrich[1]

[1]Institute of Earth Sciences, Heidelberg University, 69120 Heidelberg, Germany
[2]Institute of Geosciences, Goethe-University Frankfurt, 60438 Frankfurt, Germany

*Correspondence to*: Kim A. Jakob (kim.jakob@geow.uni-heidelberg.de)

**Abstract.** The late Pliocene/early Pleistocene intensification of Northern Hemisphere Glaciation (iNHG) ~2.5 million years ago (Marine Isotope Stages [MIS] 100–96) stands out as an important tipping point in Earth's climate history. It strongly influenced oceanographic and climatic patterns including trade-wind and upwelling strength in the Eastern Equatorial Pacific (EEP). The thermocline depth in the EEP, in turn, plays a pivotal role for the Earth's climate system: Small changes in its depth associated with short-term climate phenomena such as the El Niño-Southern Oscillation can affect surface-water properties and therefore the ocean-atmosphere exchange. However, thermocline dynamics in the EEP during the iNHG have yet remained unclear. While numerous studies have suggested a link between a thermocline shoaling in the EEP and Northern Hemisphere ice growth, other studies have indicated a stable thermocline depth during iNHG, thereby excluding a causal relationship between thermocline dynamics and ice-sheet growth. In light of these contradictory views, we have generated geochemical (planktic foraminiferal $\delta^{18}O$, $\delta^{13}C$ and Mg/Ca), sedimentological (sand-accumulation rates) and faunal (abundance data of thermocline-dwelling foraminifera) records for Ocean Drilling Program Site 849 located in the central part of the EEP. Our records span the interval from ~2.75 to 2.4 Ma (MIS G7–95), which is critical for understanding thermocline dynamics during the final phase of the iNHG. Our new records document a thermocline shoaling from ~2.64 to 2.55 Ma (MIS G2–101) and a relatively shallow thermocline from ~2.55 Ma onwards (MIS 101–95). This indicates a state change in thermocline depth at Site 849 shortly before the final phase of iNHG. Ultimately, our data support the hypothesis that (sub-)tropical thermocline shoaling may have contributed to the development of large Northern Hemisphere ice sheets.

## 1 Introduction

The onset and intensification of Northern Hemisphere Glaciation during the late Pliocene and early Pleistocene (~3.6–2.4 Ma [Mudelsee and Raymo, 2005]) is part of a long-term cooling trend following the Mid-Piacenzian warm period. The glacial corresponding to Marine Isotope Stage (MIS) G6 (~2.7 Ma) is often considered to mark the onset of large-scale glaciation in the Northern Hemisphere because it is characterized by the first occurrence of ice-rafted debris in the North

Atlantic Ocean (Bartoli et al., 2006; Bailey et al., 2013). The first culmination in Northern Hemisphere ice build-up occurred at ~2.5 Ma as documented by the first three large-amplitude (~1 ‰ in the benthic $\delta^{18}O$ record) glacial-interglacial cycles (MIS 100–96) that indicate substantial waxing and waning of ice sheets (Lisiecki and Raymo, 2005). At that time, ice rafting became widespread across the North Atlantic Ocean (Shackleton et al., 1984; Naafs et al., 2013). This so-called
"intensification of Northern Hemisphere Glaciation" (iNHG) represents an important tipping point in Earth's climate history. It strongly influenced oceanographic and climatic patterns worldwide, affecting, for example, the amount of biological production in the Eastern Equatorial Pacific Ocean (EEP) (Etourneau et al., 2010; Jakob et al., 2016) – a region that exerts a strong influence on the Earth's climate system through its effects on the global carbon and nutrient cycles (e.g., Schlitzer, 2004; Takahashi et al., 2009).

Within the EEP, the position of the thermocline in the water column bears important consequences for the Earth's climate system. Today, it is poised at shallow depths. As a consequence, small changes in its depth through east-west tilting regulated by short-term climate phenomena such as the El Niño-Southern Oscillation can affect surface-water properties and therefore ocean-atmosphere exchange processes (Fedorov et al., 2004; Ma et al., 2013). In general, proxy records and modeling results consistently document a long-term shoaling of the thermocline in the EEP and other (sub-)tropical
upwelling regions throughout the Plio-Pleistocene (Wara et al., 2005; Fedorov et al., 2006; Steph et al., 2006a, 2010; Dekens et al., 2007; Ford et al., 2012). However, the dynamics of the thermocline in the EEP and its potential links to the iNHG have yet remained enigmatic. Some studies have inferred that thermocline depth reached a critical threshold at ~3 Ma, which allowed trade winds to deliver cool waters from below the thermocline to the surface in (sub-)tropical upwelling regions such as the EEP (Fedorov et al., 2006; Dekens et al., 2007). The timing led to the hypothesis that thermocline shoaling and
the development of the EEP "cold tongue" (Wyrtki, 1981) was a necessary precondition for iNHG through reducing poleward atmospheric heat transport (Cane and Molnar, 2001). Other studies, however, identified fundamental shifts in thermocline depth only prior to ~3.5 Ma (Wara et al., 2005; Steph et al., 2006a, 2010; Ford et al., 2012), which would imply that thermocline depth in the EEP did not play an important role in the development of large-scale glaciation in the Northern Hemisphere. In light of these contradictory views, and to ultimately shed new light on potential links between low-latitude
thermocline dynamics and high-latitude ice-sheet build-up, we have investigated thermocline state changes for Ocean Drilling Program (ODP) Site 849 in the EEP during the final phase of the late Pliocene/early Pleistocene iNHG (~2.75 to 2.4 Ma, MIS G7–95).

## 2 Study area and study site

### 2.1 The Eastern Equatorial Pacific

The EEP has considerable relevance for the Earth's atmospheric and marine carbon budget (Toggweiler and Sarmiento, 1985; Takahashi et al., 2009), and at the same time exerts strong control on oceanographic and climatic circulation patterns (Fedorov and Philander, 2000; Pennington et al., 2006). Today, as a part of the tropical Pacific Walker Circulation,

westward-blowing trade winds induce a year-around upwelling of cold waters from below the thermocline to the surface in the EEP. This results in a thin, nutrient-enriched and relatively cold (~23 °C; Locarnini et al., 2013) mixed layer, the so-called EEP "cold tongue" (Wyrtki, 1981) (Fig. 1b). In today's oceans, this upwelling system supports more than 10 % of the global biological production (Pennington et al., 2006). Relatively high primary productivity rates result in exceptionally high sedimentation rates that amount to up to 3.0 cm kyr$^{-1}$ during the past 5 Myr (Mayer et al., 1992; Mix et al., 1995).

The EEP stands out as an ideal natural laboratory for studying the dynamics of the tropical thermocline: Owing to the shallow depth of the thermocline in the present-day EEP upwelling system (~50 m; Wang et al., 2000) (Fig. 1c) even small changes in its depth can affect surface-water properties such as temperature or nutrient content that can be ideally reconstructed from sediments underneath the "cold tongue". Thereby, the high sedimentation rates allow for the acquisition of proxy records at high temporal resolution compared to outside the EEP upwelling zone.

**2.2 ODP Site 849**

To reconstruct changes in thermocline depth in the EEP we focused on sediments from ODP Leg 138 Site 849 (coordinates: 0°11′N, 110°31′W; present-day water depth: 3851 m; Mayer et al., 1992) (Fig. 1a, b). This site has been selected because of (i) its position within the equatorial "cold tongue" west of the East Pacific Rise in the open ocean that makes it less prone to continental influence compared to sites east of the East Pacific Rise (Mix et al., 1995); (ii) the good preservation of foraminifera (Jakob et al., 2016, 2017) despite the fact that the present-day water depth at Site 849 is close to the lysocline (Adelseck and Anderson, 1978; Berger et al., 1982); and (iii) high sedimentation rates (2.7 cm kyr$^{-1}$ for our study interval; Jakob et al., 2017) with continuous sedimentation (Mayer et al., 1992).

**3 Investigated foraminiferal species**

The geochemical and faunal records generated in this study (for details see Section 4) are based on the planktic foraminiferal species *Globigerinoides ruber* (white, sensu stricto), *Globorotalia crassaformis*, *Globorotalia menardii*, and *Globorotalia tumida*; the presumed calcification depths of these species in the EEP as briefly elaborated upon in the following are compiled in Figure 1c. *Globigerinoides ruber* generally inhabits and calcifies in the mixed layer and is therefore typically considered to represent surface-water conditions (Fairbanks et al., 1982; Wang, 2000; Dekens et al., 2002; Steph et al., 2009, and references therein). The species *G. menardii* and *G. tumida* are found in the EEP at depths of ~25–70 m and ~50–125 m, respectively, typically in intermediate-thermocline waters (Fairbanks et al., 1982; Watkins et al., 1998; Faul et al., 2000). *Globorotalia crassaformis* inhabits the bottom of the thermocline (Niebler et al., 1999; Regenberg et al., 2009, Steph et al., 2009, and references therein) with a rather constant calcification depth as opposed to other deep-dwelling foraminiferal species (Cléroux and Lynch-Stieglitz, 2010). Although its exact calcification depth in the EEP has remained unclear, $\delta^{18}$O values of *G. crassaformis* from the (sub-)tropical Atlantic and the Caribbean Sea suggest that this species typically calcifies between ~400 and 800 m water depth (Steph et al., 2006b, 2009; Regenberg et al., 2009; Cléroux et al., 2013).

## 4 Material and methods

To reconstruct thermocline depth for the final phase of the late Pliocene/early Pleistocene iNHG from ~2.75 to 2.4 Ma (MIS G7–95), we integrate new with previously published proxy records from ODP Site 849 (Fig. 1; Tab. 1). In particular, we combine planktic (both sea-surface- and thermocline-dwelling) foraminiferal geochemical ($\delta^{18}$O, $\delta^{13}$C and Mg/Ca) proxy records with sedimentological (sand-accumulation rates) and faunal (abundance data of thermocline-dwelling foraminiferal species) information.

### 4.1 Sample material

To obtain geochemical, faunal and sedimentological information for Site 849, 374 samples have been investigated along the primary shipboard splice (Mayer et al., 1992) from cores 849C-7H-1-80 cm to 849C-7H-2-21 cm and 849D-6H-5-102 cm to 849D-7H-5-57 cm (77.02–67.78 m composite depth [mcd]). Based on the age model of Jakob et al. (2017), this interval spans from ~2.75 to 2.4 Ma (MIS G7–95). Samples with a volume of 20 cm$^3$ were investigated at 2 cm intervals, which yields a temporal resolution of ~750 yr. The sample material was dried, weighed, and washed over a 63 μm sieve.

Foraminiferal geochemical records ($\delta^{18}$O, $\delta^{13}$C and Mg/Ca) of the deep-thermocline-dwelling species *G. crassaformis* and the surface-dwelling species *G. ruber*, as well as sand-accumulation rates were generated from the full sample set (temporal resolution: ~750 yr). From these measurements surface-to-thermocline $\delta^{18}$O, $\delta^{13}$C and Mg/Ca gradients were calculated. Abundance counts of both deep-thermocline- (*G. crassaformis*) and intermediate-thermocline-dwelling species (*G. menardii* and *G. tumida*) were conducted every 20 cm (temporal resolution: ~7.5 kyr).

For *G. crassaformis*, we have generated new $\delta^{18}$O, $\delta^{13}$C and Mg/Ca records, except for the interval from ~2.65 to 2.4 Ma for which $\delta^{13}$C values have already been published by Jakob et al. (2016). For this purpose, an average of 15 individuals per sample was picked from the 315–400 μm fraction. This size fraction has been selected to keep ontogenetic effects as small as possible (Elderfield et al., 2002; Friedrich et al., 2012) but at the same time to allow for a sufficient number of *G. crassaformis* tests per sample. Moreover, the 315–400 μm fraction ensures highest comparability to previous geochemical records of this species (e.g., Karas et al., 2009; Jakob et al., 2016). Tests for $\delta^{18}$O and Mg/Ca analyses were cracked, homogenized and split into two subsamples. For *G. crassaformis*, both sinistral- and dextral-coiling specimens occur at Site 849; geochemical data were preferentially measured on tests with the most abundant coiling direction (sinistral). Sinistral-coiling specimens, however, do not occur continuously across our study interval. Thus, dextral-coiling tests were used for some intervals (in particular from ~2.46 to 2.43 Ma [69.84–69.04 mcd] and from ~2.51 to 2.50 Ma [70.85–70.68 mcd]). The occurrence of dextral- *versus* sinistral-coiling *G. crassaformis* specimens is random rather than following a specific cyclicity. Separate measurements of geochemical parameters ($\delta^{13}$C, $\delta^{18}$O, Mg/Ca) on sinistral- and dextral-coiling specimens from the same samples have demonstrated that the reconstructed values are independent of the coiling directions (see also Jakob et al., 2016).

The number of *G. crassaformis* individuals in the investigated size fraction (315–400 µm), was, however, too low to allow for geochemical analyses in specific intervals. These intervals are from 2.68 to 2.65 Ma (75.39–74.65 mcd corresponding to MIS G3) and from 2.73 to 2.69 Ma (77.02–75.79 mcd corresponding to MIS G7–G5). We decided not to fill these gaps by using another size fraction because (i) this might have biased our Mg/Ca and stable-isotope records due to ontogenetic effects (Elderfield et al., 2002; Friedrich et al., 2012), and (ii) the abundance of *G. crassaformis* specimens <315 µm was also too low for geochemical analyses in most of the samples where the 315–400 µm *G. crassaformis* size fraction is absent (see Section 5.4.2 for details).

For *G. ruber*, the previously published Mg/Ca-based sea-surface temperature (SST) and $\delta^{18}$O datasets of Jakob et al. (2017) were augmented by $\delta^{13}$C data, except for the interval from ~2.65 to 2.4 Ma for which $\delta^{13}$C values have already been published by Jakob et al. (2016). For this purpose, an average of twelve specimens was picked from the 250–315 µm size fraction (i.e., the fraction also used in Jakob et al., 2016). An overview of all geochemical datasets evaluated in this study is presented in Table 1.

### 4.2 Analytical methods

#### 4.2.1 Foraminiferal preservation

The preservation of planktic foraminiferal (*G. crassaformis* and *G. ruber*) tests used for geochemical analyses was examined by Scanning Electron Microscope (SEM) images of selected specimens from both glacial and interglacial intervals. Close-up views were taken using a LEO 440 SEM at the Institute of Earth Sciences, Heidelberg University.

#### 4.2.2 Stable-isotope analyses

Oxygen and carbon isotopes of *G. ruber* and *G. crassaformis* were analyzed using a ThermoFinnigan MAT253 gas-source mass spectrometer equipped with a Gas Bench II at the Institute of Geosciences, Goethe-University Frankfurt. Values are reported relative to the Vienna Pee Dee Belemnite (VPDB) standard through the analysis of an in-house standard calibrated to NBS-19. The precision of the $\delta^{18}$O and $\delta^{13}$C analyses is better than 0.08 ‰ and 0.06 ‰ (at 1$\sigma$ level), respectively. $\delta^{13}$C values of *G. ruber* reported herein have been adjusted for species-specific offset from equilibrium precipitation by the addition of + 0.94 ‰ (Spero et al., 2003), while measured $\delta^{18}$O values of *G. ruber* have been considered to approximate equilibrium precipitation (Koutavas and Lynch-Stieglitz, 2003) and therefore were not adjusted. For *G. crassaformis* we report the values measured on this species because laboratory investigations on $\delta^{13}$C and $\delta^{18}$O fractionation for *G. crassaformis* are still lacking.

#### 4.2.3 Mg/Ca analyses

Samples for Mg/Ca analyses of *G. crassaformis* were carefully cleaned to remove clay minerals, organic material and re-adsorbed contaminants following the protocol of Barker et al. (2003) but omitting reductive cleaning. For selected samples,

however, also reductive cleaning has been applied (see Section 5.2); as a reductive reagent a mixture of hydrazine, ammonium hydroxide and ammonium citrate was used. Analyses were carried out with an Agilent Inductively Coupled Plasma-Optical Emission Spectrometer 720 at the Institute of Earth Sciences, Heidelberg University. Reported Mg/Ca values were normalized relative to the ECRM 752-1 standard reference value of 3.762 mmol/mol (Greaves et al., 2008). To ensure instrumental precision, an internal consistency standard was monitored at least every 20 samples. Based on replicate measurements, a standard deviation for Mg/Ca of $\pm$ 0.02 mmol/mol (corresponding to $\pm$ 0.12 °C) is obtained. To identify possible contamination by clay particles or diagenetic coatings that might affect foraminiferal Mg/Ca ratios (Barker et al., 2003), elemental ratios of Al/Ca, Fe/Ca and Mn/Ca were screened (see Section 5.2).

### 4.2.4 Paleotemperature reconstruction

Species-specific conversions of Mg/Ca to temperature for *G. crassaformis* have only been calibrated based on samples from the Atlantic Ocean (Anand et al., 2003; Regenberg et al., 2009; Cléroux et al., 2013). These equations yield the same trend and amplitudes when applied to *G. crassaformis* Mg/Ca values from Site 849, but differ with regard to absolute values (Fig. 2). We converted Mg/Ca ratios of *G. crassaformis* into temperature following the species-specific equation of Cléroux et al. (2013) for two reasons: (i) The selected calibration is based on specimens with a grain size of 355–425 µm, which matches the size fraction used in our study (315–400 µm) better than the other equations that are available; and (ii) the Mg/Ca range of *G. crassaformis* from Site 849 (~1–3 mmol/mol) fits best to the calibration range (~1–2 mmol/mol) of the equation of Cléroux et al. (2013) compared to the other equations.

Because the equation of Cléroux et al. (2013) is based on oxidative and reductive cleaning of foraminiferal tests, while only oxidative cleaning has been applied to Site 849 samples, measured Mg/Ca values had to be reduced by 10 % (Barker et al., 2003). The pooled uncertainty in the temperature record is $\pm$ 0.94 °C (analytical error of $\pm$ 0.02 mmol/mol [corresponding to $\pm$ 0.12 °C] and calibrational error of $\pm$ 0.82 °C [Cléroux et al., 2013]).

### 4.2.5 Abundance counts

Following the approach of Sexton and Norris (2008), abundance counts of deep-thermocline-dwelling (*G. crassaformis,* both sinistral- and dextral-coiling specimens) and intermediate-thermocline-dwelling (*G. tumida* and *G. menardii*) species (Fig. 1c) were generated on the >250 µm size fraction of Site 849 samples. The >250 µm size fraction was split down to at least 400 individual planktic foraminifera using a microsplitter. Abundance counts were converted to mass-accumulation rates using linear sedimentation rates and dry bulk density data (calculated from high-resolution GRAPE density shipboard measurements [IODP JANUS database; Mayer et al., 1992]) following the methodology described in Jakob et al. (2016).

### 4.2.6 Sand-accumulation rates

The sand-accumulation rates available for the MIS G1–95 (~2.65–2.4 Ma) interval from Site 849 as presented in Jakob et al. (2016) were extended back to MIS G7 (~2.75 Ma) in this study. They were calculated using linear sedimentation rates, dry

bulk density data (calculated from high-resolution GRAPE density shipboard measurements [IODP JANUS database; Mayer et al., 1992]), and the portion of the >63 µm sand fraction following the approach described in Jakob et al. (2016).

## 5 Results and discussion

### 5.1 Foraminiferal test preservation at Site 849

Shipboard investigations on core catchers had originally reported a poor preservation of planktic foraminiferal tests at Site 849 (Mayer et al., 1992). However, these observations do not apply to the samples from our study interval. Instead, SEM images of both glacial and interglacial *G. crassaformis* and *G. ruber* specimens consistently show well-preserved fine features such as delicate spines, pore channels and a layered wall structure, and a lack of secondary calcite or crusts on test surfaces (Fig. 3). This indicates that test preservation is consistently sufficient for the acquisition of high-quality

geochemical data for both species throughout the study interval. A low planktic foraminiferal fragmentation index (Jakob et al., 2017) together with large numbers of well-preserved, although typically less resistant specimens such as *Globigerinoides* (Dittert et al., 1999) (Fig. 3g–l) further confirms this interpretation, indicating that dissolution has not considerably affected foraminiferal tests.

### 5.2 Assessment of contamination and diagenetic effects on Mg/Ca ratios of *G. crassaformis*

Al/Ca, Fe/Ca and Mn/Ca ratios >0.1 mmol/mol are typically considered to indicate the presence of detrital clay, Fe-rich coatings and Mn-rich overgrowth, respectively, and therefore hint at foraminiferal test contamination that might have biased measured Mg/Ca ratios (Barker et al., 2003). In most of our samples the content of Al was below the detection limit, arguing against the presence of detrital clay. With the exception of few samples, Fe/Ca values also commonly do not exceed the critical value of 0.1 mmol/mol and therefore indicate no contamination by Fe-rich overgrowth (Fig. 4a). The lack of a

statistically significant correlation between Mg/Ca and Fe/Ca ratios ($r^2 = 0.17$, p <0.01) further supports this finding.

Measured Mn/Ca ratios, however, were above the 0.1 mmol/mol threshold and therefore might indicate Mn-rich overgrowth on the analyzed tests (Fig. 4b). Reductive cleaning (in addition to the standard cleaning procedure applied to Site 849 samples; see Section 4.2.3) performed on selected samples shows by ~40–45 % lower Mn/Ca values compared to values derived from oxidative cleaning only. This indicates at least two different Mn phases coexisting on *G. crassaformis* test

surfaces. A similar observation has been made for the thermocline-dwelling species *Neogloboquadrina dutertrei* from nearby ODP Site 1240 located in the Panama Basin. In this study, Mn-rich phases not removed through oxidative cleaning are hypothesized to arise from "kutnahorite-like" carbonates or Mn-oxyhydroxide (Pena et al., 2005) – Mn phases perhaps also existing on *G. crassaformis* test surfaces at Site 849. Indeed, our data show a weak correlation between Mg/Ca and Mn/Ca ratios for *G. crassaformis* samples ($r^2 = 0.44$, p <0.01) that might indicate a bias of measured Mg/Ca ratios due to a

Mn-rich overgrowth. We note, however, that SEM images argue against the existence of any kind of overgrowth (Fig. 3). Diagenetic overgrowth usually has a Mg/Mn ratio of ~0.1 mol/mol (Barker et al., 2003, and references therein). If an

unrealistically high Mg/Mn ratio of 1 in the diagenetic overgrowth is assumed, this might change temperature estimates by on average ~0.5 °C; however, we note that this would not affect the overall shape of the Mg/Ca-based temperature record (Fig. 4c, d). We therefore conclude with reasonable certainty that early diagenetic overprinting has no significant impact on the interpretation of our data, although we cannot rule out the possibility of diagenetic changes in *G. crassaformis* tests because of significantly enriched Mn/Ca ratios.

### 5.3 Geochemical records of *G. crassaformis* and *G. ruber* at Site 849

In warm tropical waters, where seasonal climate variability is low, geochemical signatures of the investigated foraminiferal species are typically considered not to be seasonally biased (Lin et al., 1997; Tedesco et al., 2007; Mohtadi et al., 2009; Jonkers and Kučera, 2015). At Site 849, seasonal temperature variability is presently ~0.4 °C (i.e., SST of ~24 °C during summer [June] *versus* ~23.6 °C during winter [January]). The same seasonal amplitude (~0.4 °C) prevails at 20 m water depth (Locarnini et al., 2013), i.e., the assumed mean depth habitat of the here investigated species *G. ruber* (Wang, 2000; see also Section 3). Further, we suggest seasonality to decrease with increasing water depth and therefore to be <0.4 °C at the calcification depth of *G. crassaformis*. Thus, because of the low seasonal temperature variability, it is reasonable to assume that all geochemical data derived from Site 849 (Figs. 5–8) reflect mean annual conditions.

### 5.3.1 Stable-isotope data

The $\delta^{18}$O record of *G. crassaformis* varies between 0.91 ‰ and 2.79 ‰ throughout the study interval (~2.75–2.4 Ma) (Fig. 5b). Lowest values correspond to interglacials and highest values are associated with glacials. With a mean value of 2.01 ‰, $\delta^{18}$O values of *G. crassaformis* (this study) are on average 2.78 ‰ higher than those of *G. ruber* (Jakob et al., 2017), indicating cooler and/or more saline waters at the calcification depth of *G. crassaformis* (bottom of the thermocline) compared to that of *G. ruber* (surface waters) (Fig. 1c; see Section 3 for details). In theory, a deeper calcification depth of *G. crassaformis* than of *G. ruber* should result in a lower $\delta^{13}$C signature in *G. crassaformis* tests than in those from *G. ruber* since the amount of organic matter remineralization and the associated release of light $^{12}$C into the surrounding water typically increases with water depth (Deuser and Hunt, 1969; Kroopnick, 1985). Indeed, by ~1.86 ‰ lower $\delta^{13}$C values are recorded for *G. crassaformis* (mean: 0.46 ‰) than for *G. ruber* (mean: 2.32 ‰; Jakob et al., 2016; this study) (Fig. 5c). In general, $\delta^{13}$C values of *G. ruber* and *G. crassaformis* fluctuate between minima of 1.36 ‰ and 0 ‰ during glacials and maxima of 2.97 ‰ and 0.97 ‰ during interglacials, respectively (Jakob et al., 2016; this study) (Fig. 5c). It has been shown in a former study that the glacial-interglacial foraminiferal $\delta^{13}$C pattern at Site 849 during iNHG is strongly controlled by the amount of primary productivity in the Southern Ocean, which is highest during interglacials (high $\delta^{13}$C) and lowest during glacials (low $\delta^{13}$C) (Jakob et al., 2016).

**5.3.2 Mg/Ca-based temperature data**

Mg/Ca values of *G. crassaformis* vary between 0.82 mmol/mol and 3.02 mmol/mol (this study), being on average 1.43 mmol/mol lower than Mg/Ca ratios of *G. ruber* from the same site and time interval (Jakob et al., 2017) (Fig. 5d). Temperatures reconstructed from *G. crassaformis* Mg/Ca values cover a range from 1.0 °C to 11.6 °C. Thus, sub-thermocline temperatures at Site 849 derived from *G. crassaformis* are on average 19 °C lower than SSTs as reflected by *G. ruber* (~22–27 °C; Jakob et al., 2017) (Fig. 5d). To bring *G. crassaformis*-based temperatures of this study into a broader temporal and geographical context we next compare this record to a number of other relevant datasets:

(i) The temperature range reconstructed based on *G. crassaformis* at Site 849 is close to temperatures inferred for the same species and time interval at Deep Sea Drilling Project (DSDP) Site 214 in the tropical eastern Indian Ocean (~8–10 °C; Karas et al., 2009) (Figs. 1a, 6b). Yet available Mg/Ca-based thermocline-temperature records from the east Pacific for the ~2.75–2.4 Ma interval capture ranges of ~17–22.5 °C (ODP Site 1241, *G. tumida*-based; Steph et al., 2006a), ~18–21 °C (Site 1241, *N. dutertrei*-based; Groeneveld et al., 2014) and ~15.5–17.5 °C (Site 849, *G. tumida*-based; Ford et al., 2012) (Figs. 1a, b and 6b, c). In comparison to thermocline temperature records from Site 1241 (Steph et al., 2006a; Groeneveld et al., 2014) our new *G. crassaformis*-based temperatures appear to be relatively low. However, this discrepancy can likely be explained with different calcification depths of the investigated species (and therefore different temperatures reflected; Cléroux et al., 2013) together with differences in the oceanographic setting of the investigated sites (i.e., Site 849 in the "cold tongue" *versus* Site 1241 in the "warm pool" with, for example, a modern-day SST difference of ~3 °C; Locarnini et al., 2013; see also Section 5.4.1). The record from Site 849 by Ford et al. (2012) is of low temporal resolution with only six datapoints across our study interval such that peak (both glacial and interglacial) temperatures are probably not captured. A comparison to this record is therefore not robust enough to warrant further discussion.

(ii) The EEP upwelling system is mainly fed by waters derived from the Southern Ocean (e.g., Tsuchiya et al., 1989). However, relatively warm SSTs in the Southern Ocean during our study interval (~10.5–17 °C at ODP Site 1090 [Martínez-Garcia et al., 2010; Fig. 1a] – a site accepted as the best endmember of Southern Ocean waters [Billups et al., 2002; Pusz et al., 2011]) are difficult to reconcile with substantially lower temperatures recorded in thermocline waters at EEP Site 849 (~1–11.6 °C; this study) and Indian Ocean Site 214 (~8–10 °C; Karas et al., 2009). Together, these datasets imply strong cooling of Southern Ocean surface waters either when being downwelled and transported to the lower latitudes and/or through mixing with cold (presently ~0 °C; Craig and Gordon, 1965) Antarctic Bottom Waters.

(iii) Thermocline temperatures of the Last Glacial Maximum are expected to be substantially colder than those of the Plio-/Pleistocene transition (Ford et al., 2012). However, in comparison to tropical east Pacific thermocline temperatures of the Last Glacial Maximum (on average ~14–16 °C) that derive from Mg/Ca data of *G. tumida* from Site 849 (Ford et al., 2015) and *N. dutertrei* from Sites 08JC and 17JC (Hertzberg et al., 2016) (Fig. 1a) deep-thermocline temperatures of *G. crassaformis* at Site 849 from the ~2.75–2.4 Ma interval appear to be too low. We suggest that different calcification depths of the investigated species may explain this discrepancy (Cléroux et al., 2013).

(iv) Finally, we notice that modern bottom-water temperatures at our study site of ~1.5 °C (Locarnini et al., 2013) overlap the deep-thermocline temperature range as reconstructed from *G. crassaformis*-based Mg/Ca data. Very low *G. crassaformis*-based temperatures of ~1 °C derive, however, from a single datapoint, while temperatures typically higher than 2–3 °C even during prominent iNHG glacials can be reconciled better with present-day bottom-water temperatures.

## 5.4 Thermocline development in the Eastern Equatorial Pacific

### 5.4.1 Geochemical evidence

Variations in the vertical temperature gradient within the upper water column allow to effectively monitor shifts in thermocline depth with a small temperature difference between surface and thermocline waters indicating a deep thermocline and *vice versa* (e.g., Steph et al., 2009; Nürnberg et al., 2015). Absolute values of surface-to-thermocline temperature gradients can be reconstructed from the Mg/Ca-based temperature gradient derived from surface-dwelling and thermocline-dwelling species. In addition, the $\delta^{18}O$ gradient provides information on relative changes in the surface-to-thermocline temperature gradient (as opposed to absolute estimates derived from the Mg/Ca-gradient). This is because salinity might also play a role in controlling $\delta^{18}O$ values of foraminifera inhabiting different depths. The effect of global ice volume (i.e., the third factor beside temperature and salinity that affect foraminiferal $\delta^{18}O$; Ravelo and Hillaire-Marcel, 2007) should be identical. For gradient calculations the species *G. ruber* is ideally suited as a surface-water recorder since it is one of the shallowest-dwelling species among modern planktic foraminifera (Bé, 1977), while *G. crassaformis* appears to be one of the most promising recorder of the deep thermocline because of its rather constant calcification depth compared to other deep dwellers (Cléroux and Lynch-Stieglitz, 2010). This approach (i.e., *G. ruber* to *G. crassaformis* gradient calculation) has already successfully been applied by previous studies that have investigated changes in surface-water structure and thermocline through time (e.g., Karas et al., 2009; Bahr et al., 2011).

The $\delta^{18}O$- and Mg/Ca-based temperature gradients between *G. ruber* and *G. crassaformis* at Site 849 show no glacial-interglacial cyclicity (Fig. 7b), indicating that thermocline depth was unaffected by varying glacial *versus* interglacial climatic conditions. This observation is in line with previous modeling efforts (Lee and Poulsen, 2005) and proxy-based (both geochemical and faunal) studies from EEP "cold tongue" ODP Sites 846 and 849 that cover the same time interval (Bolton et al., 2010; Jakob et al., 2017). However, it contradicts geochemical data (i.e., the surface-to-thermocline [*Globigerinoides sacculifer* to *N. dutertrei*] Mg/Ca-based temperature gradient) derived from Site 1241 located in the east Pacific "warm pool" that shows glacial-interglacial variations in thermocline depth for MIS 100–96 (Groeneveld et al., 2014) (Fig. 8b). Therefore, we hypothesize that thermocline depth underwent a different evolution on the glacial-interglacial timescale in regions inside (Sites 846 and 849) and outside (Site 1241) the "cold tongue". Further, water-column stratification and therefore thermocline depth at Site 1241 has been shown to be affected by glacioeustatic closures of the Central American Seaway that temporally occurred during MIS 100–96 (Groeneveld et al., 2014), perhaps through upwelling intensification (Schneider and Schmittner, 2006). In contrast, Site 849 appears to be unaffected by such changes as

shown by sand-accumulation-rate-based data on export production (Jakob et al., 2016). Hence it is likely that Site 1241 reflects a more local signal as opposed to an open-ocean, quasi-global signal reported at Site 849 due to its position west of the East Pacific Rise (Mix et al., 1995).

On longer timescales, the $\delta^{18}$O gradient at Site 849 decreased by ~1 ‰ from ~2.64 Ma to 2.55 Ma (MIS G2–101),

and remained relatively constant throughout the remainder of our study interval (~2.55–2.38 Ma; MIS 101–95) (Fig. 7b). The Mg/Ca-based temperature gradient shows the same overall pattern as the $\delta^{18}$O gradient (i.e., a ~5 °C increase from ~2.64 Ma to 2.55 Ma, and a constant value of ~20 °C from ~2.55 to 2.38 Ma) (Fig. 7b). We interpret the decrease in the $\delta^{18}$O gradient and the increase in the Mg/Ca-based temperature gradient from ~2.64 Ma to 2.55 Ma to represent a shoaling of the thermocline. From that time onwards, constant $\delta^{18}$O and Mg/Ca-based temperature gradients suggest that the thermocline

remained relatively shallow throughout the final phase of the late Pliocene/early Pleistocene iNHG until ~2.38 Ma. In general, the shoaling trend of the thermocline revealed by our records agrees with the overall long-term shoaling trend of the thermocline in the EEP observed throughout the Plio-Pleistocene (Fedorov et al., 2006; Steph et al., 2006a, 2010; Dekens et al., 2007; Ford et al., 2012), although substantial thermocline shoaling is restricted to the ~2.64–2.55 Ma period rather than occurring continuously throughout the entire investigated time interval (Fig. 8). However, our data do not confirm previous

datasets indicating that most prominent thermocline shoaling occurred prior to ~3.5 Ma (Wara et al., 2005; Steph et al., 2006a, 2010; Ford et al., 2012). Geochemical data from Site 1241 even reveal a transient thermocline deepening for the ~2.7–2.5 Ma interval (Steph et al., 2006a) (Fig. 8b), which also cannot be confirmed by our records from Site 849 and therefore supports that Site 1241 might reflect a more local signal (see above). Other records used to infer thermocline evolution in the east Pacific for this interval lack the required temporal resolution (~10–30 kyr [Wara et al., 2005; Dekens et

al., 2007; Ford et al., 2012] *versus* ~0.75–3 kyr [Steph et al., 2006a; this study]) to resolve transient thermocline changes as observed at Site 1241, if existent.

### 5.4.2 Faunal and sedimentological evidence

Based on our data, the overall abundances of the deep-thermocline-dwelling species *G. crassaformis* at Site 849 were relatively low when the thermocline was relatively deep (i.e., prior to ~2.55 Ma as indicated by the $\delta^{18}$O and Mg/Ca-based

temperature gradients) (Fig. 7c). More specifically, *G. crassaformis* was present in substantially reduced numbers only or even completely absent between ~2.73 and 2.64 Ma (MIS G7–G2; with the exception of MIS G4 [~2.69–2.68 Ma]). Since ~2.64 Ma, i.e., when the thermocline started to shoal, relative *G. crassaformis* abundances increased markedly from typically <5 % to ~10–35 % such that representatives of this taxon are present continuously throughout the remainder of the study interval. The increase of *G. crassaformis* abundances occurs at the expense of the intermediate-thermocline-dwelling species

*G. menardii* and *G. tumida*, which decline from ~30–75 % to only ~5–65 % (Fig. 7c). This observation suggests that prior to the final phase of the late Pliocene/early Pleistocene iNHG, when the thermocline was relatively deep, the living conditions for the deep-thermocline-dwelling species *G. crassaformis* were unfavorable. At the same time, low *G. crassaformis* abundances allowed intermediate-thermocline-dwelling species (such as *G. menardii* and *G. tumida*) to dominate the

planktic foraminiferal assemblages. However, we note that thermocline depth is most probably not the sole factor regulating *G. crassaformis* abundances. This is because this species reaches only low percentages within modern planktic foraminiferal assemblages in the EEP (typically <1 % in core-top samples; Prell 1985) although the present-day thermocline is relatively shallow.

In general, *G. crassaformis* reaches highest abundances in oxygen-depleted waters (Jones, 1967; Kemle von Mücke and Hemleben, 1999), while other environmental factors including temperature, salinity or nutrient availability are markedly less important (Cléroux et al., 2013). The oxygen content of deep waters (and accordingly the abundance of *G. crassaformis*) is typically closely coupled to surface-water productivity (with a lower oxygen level and higher *G. crassaformis* abundances in deeper waters when surface-water productivity is high) (Wilson et al., 2017). We therefore hypothesize that primary productivity at Site 849 prior to ~2.64 Ma (MIS G2) was relatively low (with the exception of MIS G4). Low primary productivity led to a reduction of organic matter remineralization and therefore oxygen consumption in deeper waters. An elevated oxygen content in deeper waters implies, in turn, low *G. crassaformis* abundances. Such a scenario is supported by the sand-accumulation-rate-based primary productivity record from Site 849 (Jakob et al., 2016; this study), which clearly indicates low productivity rates prior to ~2.64 Ma when the thermocline was relatively deep compared to after ~2.64 Ma when the thermocline became shallow (Fig. 7b, d).

As outlined above, on longer timescales primary productivity at Site 849 appears to be coupled to variations in thermocline depth. The observed long-term trend in primary productivity is overprinted by a clear glacial-interglacial cyclicity particularly during the final phase of iNHG (MIS 100–96); the position of the thermocline, however, remained constant along the obliquity (i.e., 41-kyr) band (Fig. 7b, d). Notably, the accumulation rates of *G. crassaformis, G. menardii* and *G. tumida* perfectly capture the pattern derived by the sand-accumulation-rate-based productivity record (Fig. 7c, d). This suggests that the abundances of deep- and intermediate-thermocline-dwelling foraminiferal species are coupled to the strength of the biological pump and can be used as a tracer for primary productivity at Site 849.

The relatively high $\delta^{13}C$ gradient for MIS G4–G2 as it emerges from our data (Fig. 7d) confirms the interpretation of Jakob et al. (2016) for MIS G1–95 that low-amplitude productivity changes prior to MIS 100 were mainly driven by the nutrient content within the upwelled water mass as long as the thermocline was relatively deep (as reflected by a low thermocline-to-surface temperature gradient prior to ~2.55 Ma [MIS 101]; Fig. 7b, d). High-amplitude productivity changes resulting from upwelling intensification played an important role from MIS 100 onward (Jakob et al., 2016). At the same time, the thermocline became relatively shallow (as reflected by a large thermocline-to-surface temperature gradient since ~2.55 Ma [MIS 101] at Site 849; Fig. 7b, d). This is unlikely a coincidence, but rather suggests that the thermocline depth reached a critical threshold at that time. Prior to this, relatively warm and nutrient-poor waters from above the thermocline have upwelled at EEP Site 849. Consequently, primary productivity rates were low. When the thermocline became shallow enough, however, trade winds could deliver cooler, nutrient-enriched waters from below the thermocline to the surface, which allowed primary productivity to increase. Such a scenario is supported by the observation that lower SSTs prevailed at Site 849 during glacials since MIS 100 (Jakob et al., 2017) (Fig. 5d). Moreover, temporal closures of the Central American

Seaway might have additionally promoted generally increasing upwelling and therefore primary productivity rates since MIS 100 (Schneider and Schmittner, 2006; Groeneveld et al., 2014).

Our findings imply a marked effect of long-term thermocline state change on primary productivity at the studied site. In particular, when thermocline shoaling reached a critical threshold, primary productivity increased, thereby removing $CO_2$ from the surface ocean. Thermocline shoaling appears to be a consistent feature in the EEP throughout the Plio-/Pleistocene (although the timing and magnitude varies between different studies which may be related to different approaches for thermocline reconstruction or different localities and therefore local oceanography). Considering that the entire EEP contributes substantially to global biological production in the present-day oceans (Pennington et al., 2006), the observed coupling between thermocline state change and primary productivity (if behaving similar in the entire EEP "cold tongue") is of major importance for the Earth's climate. It may have favored global cooling and therefore the early development of large ice sheets in the Northern Hemisphere. At the same time, as suggested in previous studies, the formation of the "cold tongue" through thermocline shoaling might have additionally preconditioned the iNHG by reducing atmospheric heat transport from the tropics to the poles (Cane and Molnar, 2001).

## 6 Conclusions

We integrate new with previously published foraminiferal-based geochemical, faunal, and sedimentological records for ODP Site 849 (~2.75–2.4 Ma, MIS G7–95) to reconstruct changes in thermocline depth during the late Pliocene/early Pleistocene iNHG. Our data document a shoaling of the thermocline at Site 849 from ~2.64 to 2.55 Ma, while it remained relatively shallow until ~2.38 Ma, implying that major changes in thermocline depth occurred shortly before the final phase of the late Pliocene/early Pleistocene iNHG (i.e., prior to MIS 100–96). This finding, which is in line with former studies (Fedorov et al., 2006; Dekens et al., 2007), supports the hypothesis that (sub-)tropical thermocline shoaling was a precondition to allow the development of large ice sheets in the Northern Hemisphere (Cane and Molnar, 2001). At the same time, our new data contradict studies that have documented substantial shifts in thermocline depth in the EEP only prior to ~3.5 Ma (Wara et al., 2005; Steph et al., 2006a, 2010; Ford et al., 2012). Our new records also suggest low primary productivity rates during times of a relatively deep thermocline prior to ~2.64 Ma (MIS G2). In turn, the relatively shallow thermocline associated with low SSTs after ~2.55 Ma allowed primary productivity to increase during prominent iNHG glacials (MIS 100–96) and, perhaps, by removing $CO_2$ from ocean-atmosphere exchange processes, to further stimulate Northern Hemisphere ice-sheet growth.

## Data availability

All data reported will be made available upon publication of this paper via the open access PANGAEA database (www.pangaea.de).

**Author contributions**

The project was designed by OF, KAJ, and JP. KAJ and CS carried out Mg/Ca analyses; JF and KAJ performed stable-isotope analyses. KAJ wrote the manuscript with input from all co-authors.

**Competing interests**

The authors declare that they have no conflict of interest.

**Acknowledgments**

Richard Norris provided invaluable support in foraminiferal taxonomy. Sven Hofmann and Silvia Rheinberger are thanked for stable-isotope and Mg/Ca analyses, respectively. André Bahr provided support in Mg/Ca sample preparation. Hans-Peter Meyer and Alexander Varychev provided SEM assistance. Jani L. Biber, Verena Braun, Jakob Gänzler, Barbara Hennrich,

Karsten Kähler, and Tobias Syla helped with processing of sediment samples. Comments and suggestions by editor Luc Beaufort and two anonymous reviewers are highly appreciated. This research used samples provided by the Ocean Drilling Program, which was sponsored by the U.S. National Science Foundation and participating countries under the management of Joint Oceanographic Institutions, Inc. Funding for this study was provided by the German Research Foundation (DFG; grants FR2544/6 to O.F. and PR651/15 to J.P.).

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

**Table 1:** Compilation of the geochemical datasets from ODP Site 849 evaluated in this study. The number of samples indicates the total amount of samples processed (dried, weighed, washed) during each study, while the number of geochemical datapoints is somewhat lower depending on the availability of foraminiferal (*G. crassaformis* and *G. ruber*) material in each sample.

| Foraminiferal species | Size fraction | Proxy | Interval | No. of samples (datapoints) | Reference |
|---|---|---|---|---|---|
| *G. crassaformis* (sinistral- and dextral-coiling) | 315–400 µm | $\delta^{13}$C | 74.17–67.78 mcd (~2.65–2.4 Ma; MIS G1–95) | 229 (215) | Jakob et al. (2016) |
| | | | 77.02–74.19 mcd (~2.75–2.65 Ma; MIS G7–G2) | 145 (43) | this study |
| | 315–400 µm | $\delta^{18}$O | 77.02–67.78 mcd (~2.75–2.4 Ma; MIS G7–95) | 374 (258) | this study |
| | 315–400 µm | Mg/Ca | 77.02–67.78 mcd (~2.75–2.4 Ma; MIS G7–95) | 374 (247) | this study |
| *G. ruber* (white, sensu stricto) | 250–315 µm | $\delta^{13}$C | 74.17–67.78 mcd (~2.65–2.4 Ma; MIS G1–95) | 229 (225) | Jakob et al. (2016) |
| | | | 77.02–74.19 mcd (~2.75–2.65 Ma; MIS G7–G2) | 145 (137) | this study |
| | 250–315 µm | $\delta^{18}$O | 77.02–67.78 mcd (~2.75–2.4 Ma; MIS G7–95) | 374 (362) | Jakob et al. (2017) |
| | 200–250 µm | Mg/Ca | 77.02–67.78 mcd (~2.75–2.4 Ma; MIS G7–95) | 374 (316) | Jakob et al. (2017) |

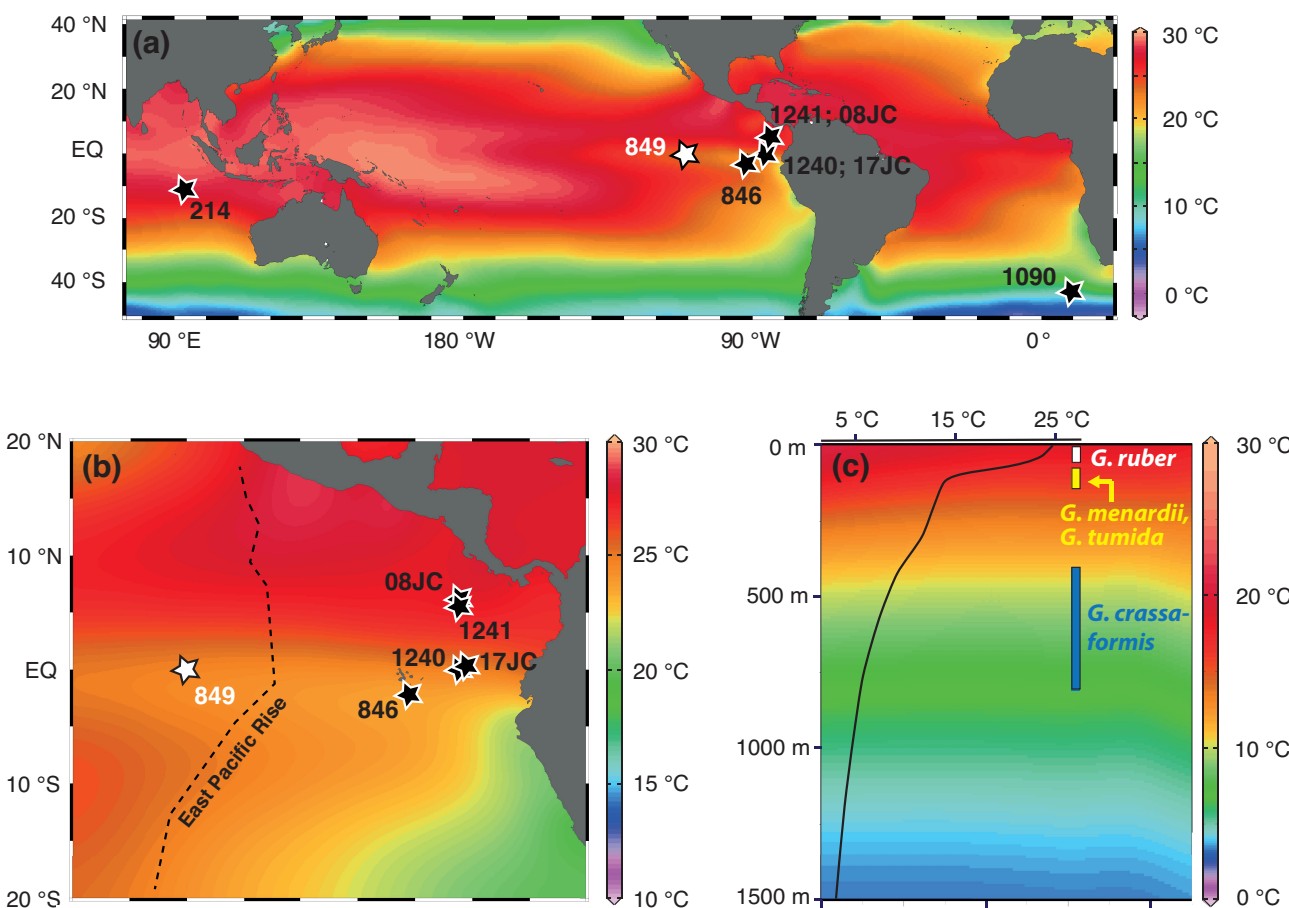

**Figure 1: Location of study area and present-day temperature profile at studied site.** (a) Map indicating the location of the study site (ODP Site 849; white) and other sites mentioned in the text (black). (b) Map showing the location of Site 849 in the EEP "cold tongue" (white) and other east Pacific sites mentioned in the text (black). Colors in (a) and (b) denote mean annual surface-water temperatures. (c) Mean annual temperature profile for Site 849 showing the position of the thermocline. Calcification depths of foraminiferal species analyzed in this study are indicated by white (*G. ruber*), yellow (*G. menardii* and *G. tumida*) and blue (*G. crassaformis*) bars (because the calcification depth of *G. crassaformis* in the EEP remains unclear, we here show its calcification depth range identified for the [sub-]tropical Atlantic and the Caribbean Sea; see Section 3 for details). Maps are after World Ocean Atlas (Locarnini et al., 2013).

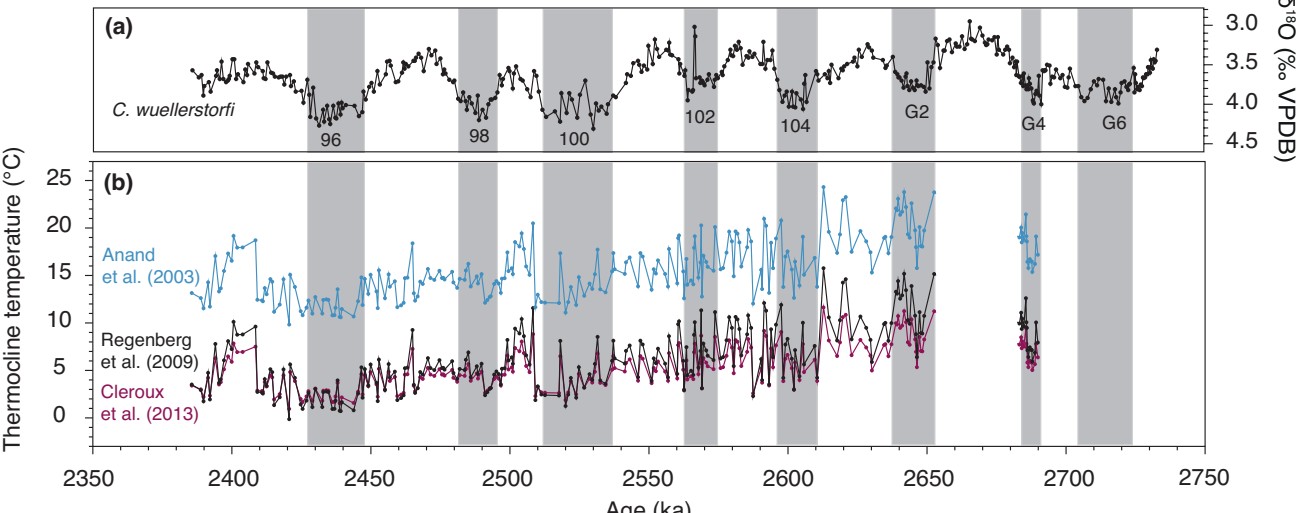

**Figure 2: Species-specific conversions of Mg/Ca ratios to temperature for *G. crassaformis*.** (a) Benthic foraminiferal (*C. wuellerstorfi*) $\delta^{18}$O stratigraphy (Jakob et al., 2017). (b) Comparison of the different species-specific Mg/Ca-to-temperature conversions of Cléroux et al. (2013; red), Regenberg et al. (2009; black) and Anand et al. (2003; blue) applied to *G. crassaformis* Mg/Ca values from Site 849 for ~2.75 to 2.4 Ma. Gaps in the dataset for 2.68–2.65 Ma (MIS G3) and 2.73–2.69 Ma (MIS G7–G5) are due to a lack of *G. crassaformis* specimens in the investigated (315–400 μm) size fraction. Grey bars highlight glacial periods.

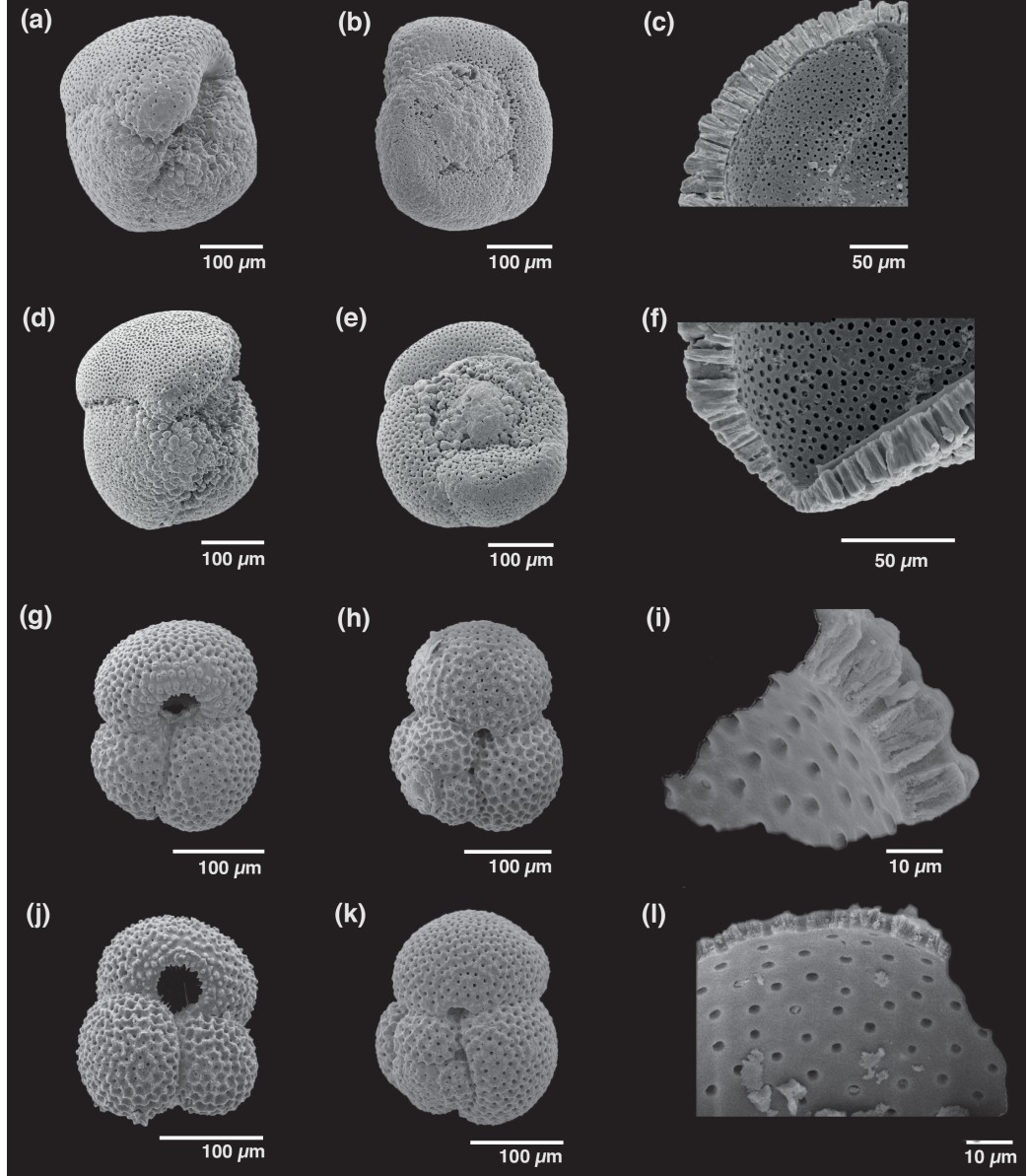

**Figure 3: Scanning Electron Micrographs of *Globorotalia crassaformis* and *Globigerinoides ruber* from Site 849.** Both glacial (a–c, g–i) and interglacial (d–f, j–l) foraminiferal tests are well preserved, allowing for the acquisition of reliable geochemical data.

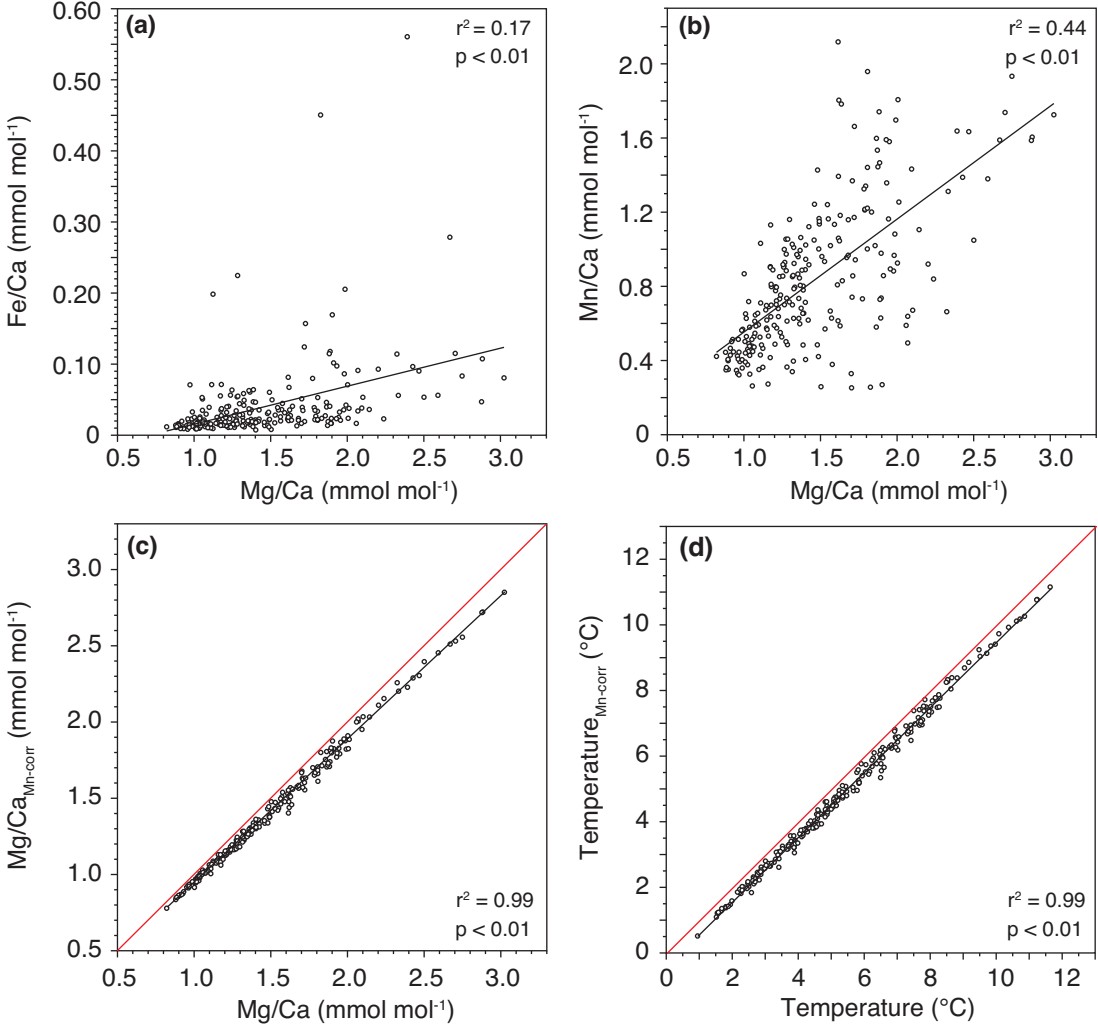

**Figure 4: Evaluation of potential contamination on Mg/Ca ratios in *G. crassaformis*.** (a) Cross plot between measured Mg/Ca and Fe/Ca ratios. (b) Cross plot between measured Mg/Ca and Mn/Ca ratios. (c) Cross plot between measured Mg/Ca and Mg/Ca ratios corrected for Mn-bearing overgrowths (using the assumption of an unrealistically high Mg/Mn ratio of 1) in comparison to uncontaminated samples (red line). (d) Cross plot between temperatures calculated from measured Mg/Ca and Mg/Ca ratios corrected for Mn-bearing overgrowths (using the assumption of an unrealistically high Mg/Mn ratio of 1) in comparison to uncontaminated samples (red line) (see Section 5.2 for details).

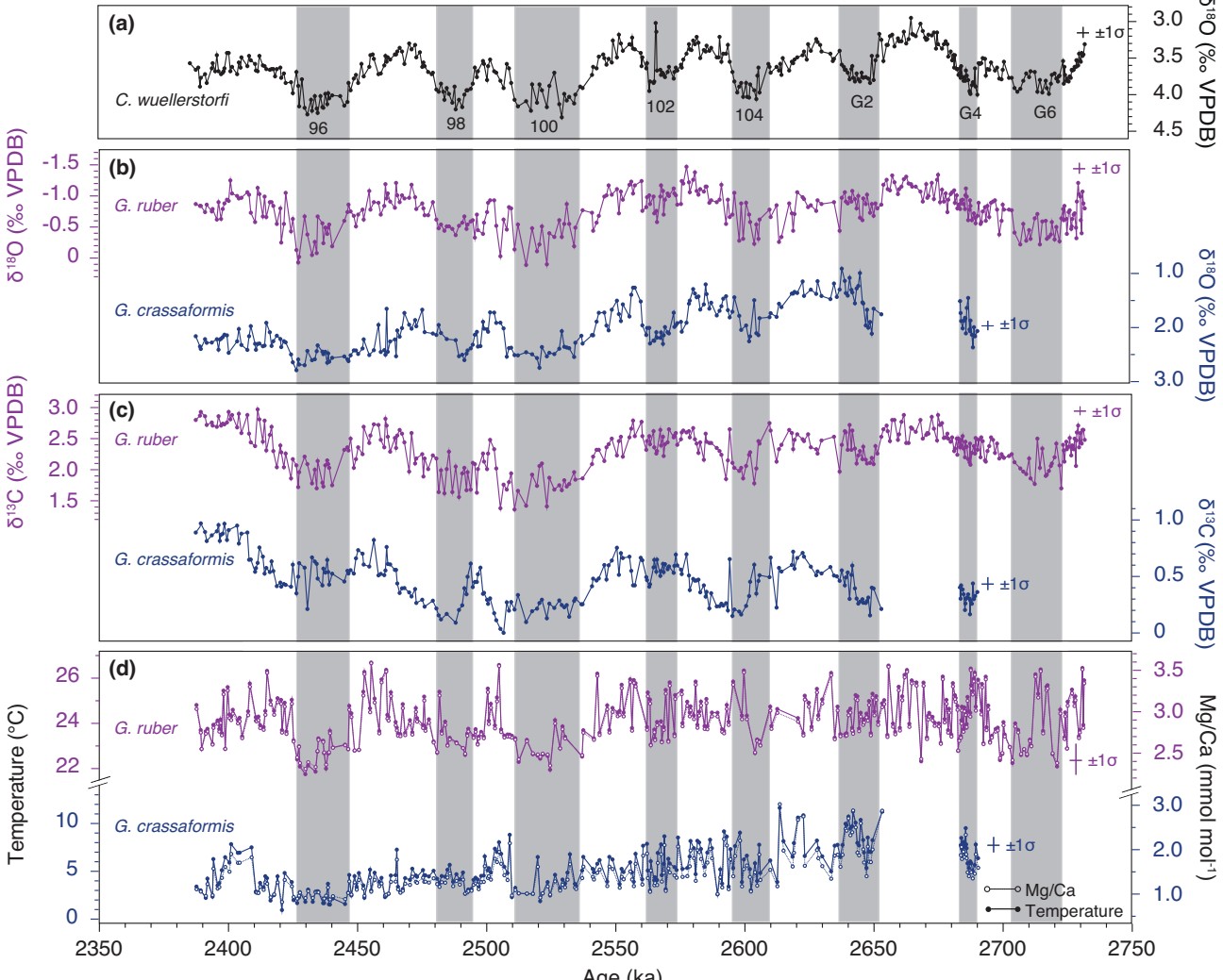

**Figure 5: Planktic foraminiferal proxy records from ODP Site 849 for MIS G7–95 (~2.75–2.4 Ma).** (a) Benthic foraminiferal (*C. wuellerstorfi*) δ¹⁸O stratigraphy (Jakob et al., 2017). (b) δ¹⁸O data of *G. ruber* (purple; Jakob et al., 2017) and *G. crassaformis* (blue; this study). (c) δ¹³C data of *G. ruber* (purple; Jakob et al., 2016, this study) and *G. crassaformis* (blue; Jakob et al., 2016, this study). (d) Mg/Ca (dashed line, open dots) and temperature (solid line, filled dotes) data of *G. ruber* (purple; Jakob et al., 2017) and *G. crassaformis* (blue; this study). Gaps in the *G. crassaformis* dataset for 2.68–2.65 Ma (MIS G3) and 2.73–2.69 Ma (MIS G7–G5) are due to a lack of *G. crassaformis* specimens in the investigated (315–400 µm) size fraction. Horizontal and vertical bars indicate the 1σ standard deviation associated with the age model and the individual proxy records, respectively. Grey bars highlight glacial periods.

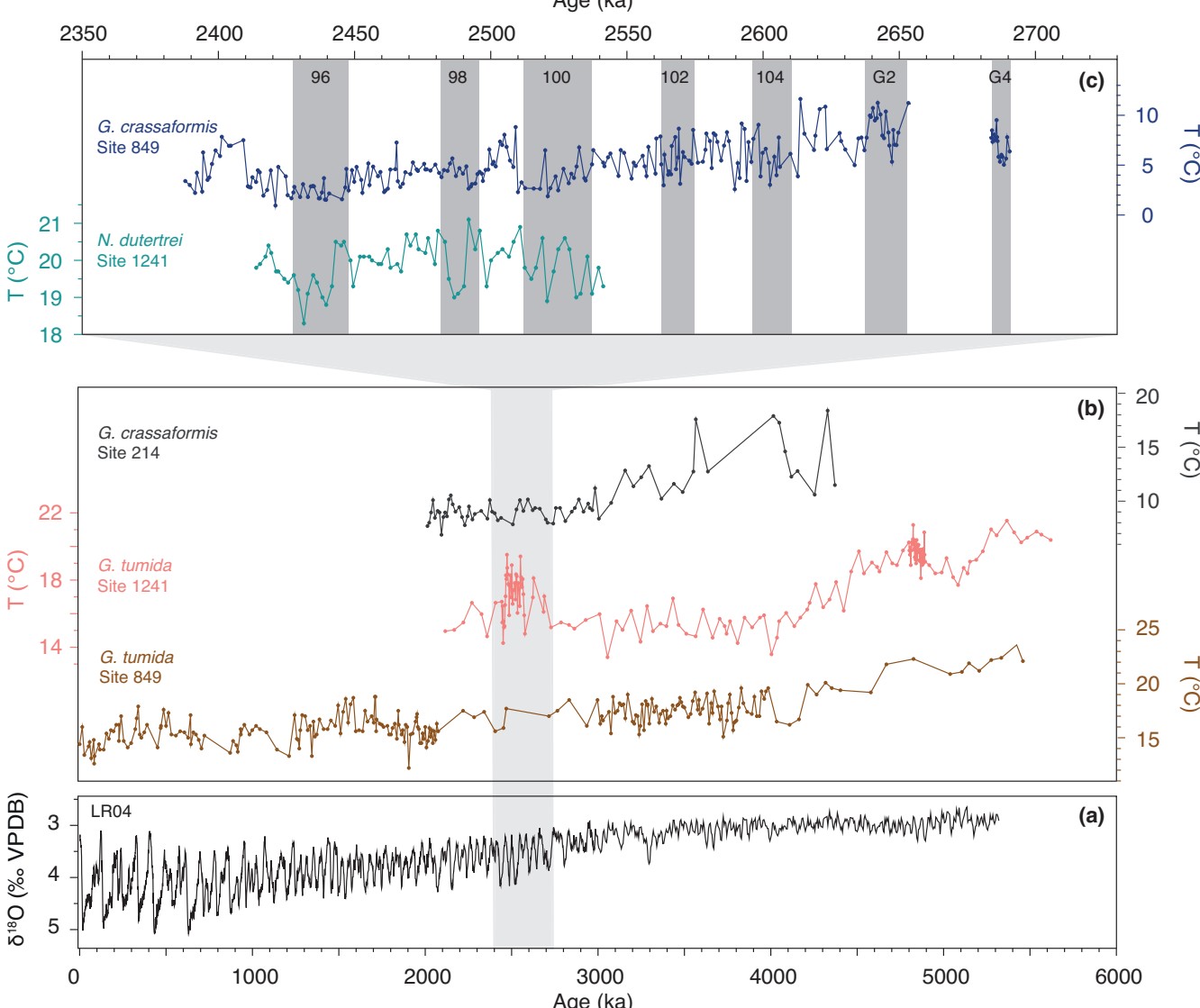

**Figure 6: Comparison of selected thermocline-temperature records from the tropical Pacific for the past ~6 Myr (bottom) and the ~2.75–2.4 Ma interval (top).** (a) Benthic oxygen isotope record (LR04 stack; Lisiecki and Raymo, 2005). (b) Low-resolution Mg/Ca-based temperature records of thermocline-dwelling species from Site 214 (grey; Karas et al., 2009), Site 1241 (pink; Steph et al., 2006a) and Site 849 (brown; Ford et al., 2012). Light grey shading in a and b marks the investigated time period of this study. (c) High-resolution Mg/Ca-based temperature records of thermocline-dwelling species from Site 849 (blue; this study) and Site 1241 (cyan; Groeneveld et al., 2014). Grey bars highlight glacial periods.

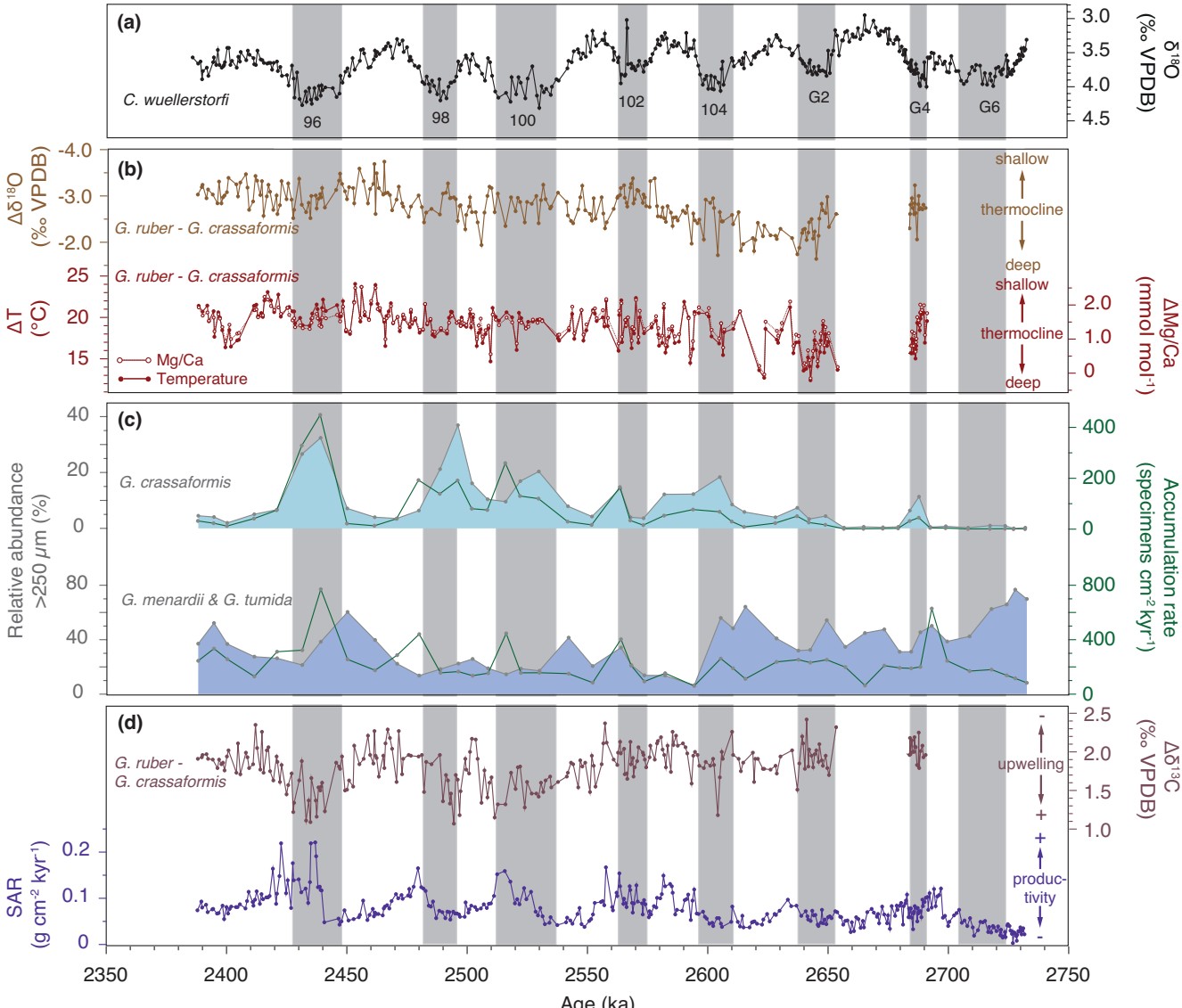

**Figure 7: Compilation of planktic foraminiferal, faunal and sedimentary proxy records from ODP Site 849 for MIS G7–95 (~2.75–2.4 Ma).** (a) Benthic foraminiferal (*C. wuellerstorfi*) $\delta^{18}$O stratigraphy (Jakob et al., 2017). (b) $\delta^{18}$O (brown; this study) and Mg/Ca (red dashed line with open dots; this study) and temperature (red; this study) gradients between the surface-dwelling species *G. ruber* and the thermocline-dwelling species *G. crassaformis* as a proxy for relative thermocline depth. (c) Relative abundances of *G. crassaformis* (light blue; this study) and *G. menardii* and *G. tumida* (dark blue; this study) together with their mass-accumulation rates (green; this study). (d) $\delta^{13}$C gradient between *G. ruber* and *G. crassaformis* as an indicator for upwelling strength (purple; Jakob et al., 2016, this study) together with sand-accumulation rates (SAR) as an indicator for primary productivity (Jakob et al., 2016, this study). Gaps in the $\delta^{18}$O, $\delta^{13}$C and Mg/Ca gradients for 2.68–2.65 Ma (MIS G3) and 2.73–2.69 Ma (MIS G7–G5) are due to a lack of *G. crassaformis* specimens in the investigated (315–400 µm) size fraction. Grey bars highlight glacial periods.

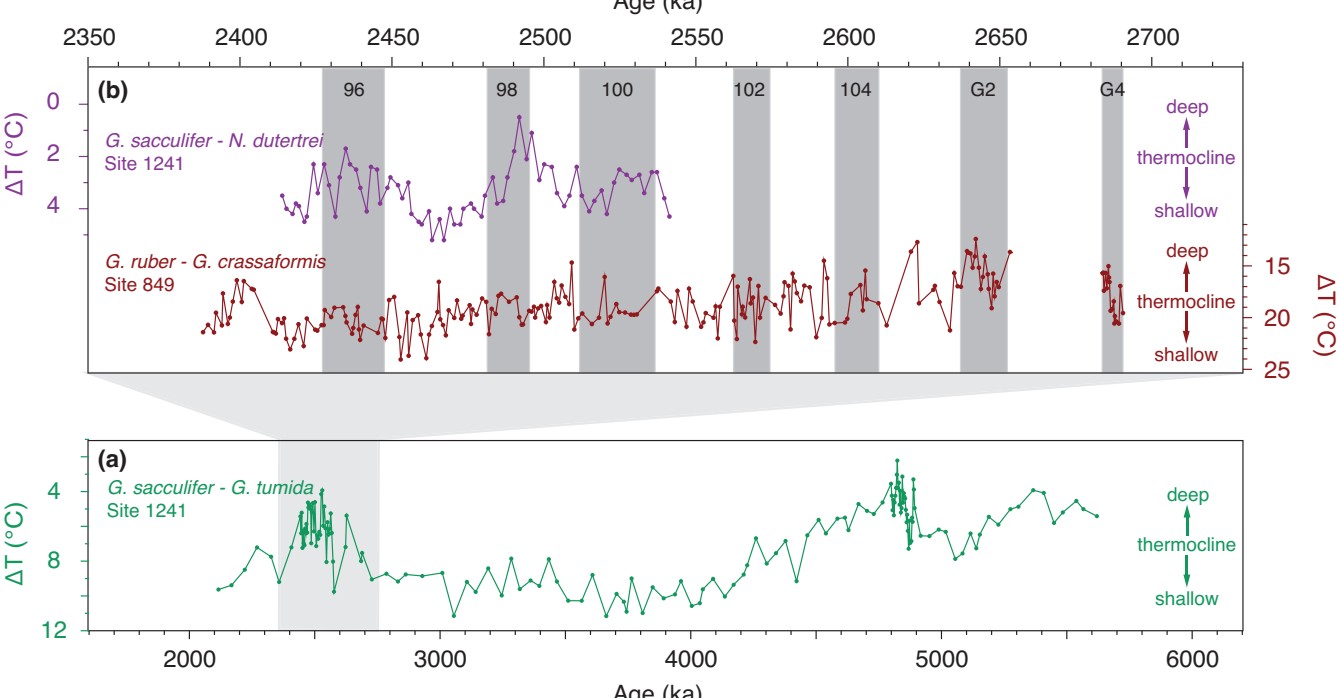

**Figure 8: Comparison of selected thermocline-depth proxy records from the east Pacific for the ~6–2 Ma (bottom) and the ~2.75–2.4 Ma (top) intervals.** (a) Low-resolution surface-to-thermocline temperature gradient from Site 1241 (green; Groeneveld et al., 2006; Steph et al., 2006a) reflecting relative thermocline depth changes. Light grey shading marks the investigated time period of this study. (b) High-resolution surface-to-thermocline temperature gradients from Site 849 (red; this study) and Site 1241 (purple; Groeneveld et al., 2014) indicative of relative thermocline depth changes. Grey bars highlight glacial periods.