# Peer review of "Thermocline state change in the Eastern Equatorial Pacific during the late Pliocene/early Pleistocene intensification of Northern Hemisphere Glaciation"

_Climate of the Past, 2017_

## Referee Comment (RC1) · Anonymous Referee #1 · 18 Jan 2018

Review of the manuscript "Thermocline state change in the Eastern Equatorial Pacific during the late Pliocene/early Pleistocene intensification of Northern Hemisphere Glaciation" by Jakob et al. This manuscript uses a combination of foraminiferal geochemistry (Mg/Ca and stable oxygen and carbon isotopes), abundance and sand accumulation rate to reconstruct how the thermocline in the east Pacific cold tongue area developed during the onset of Northern Hemisphere Glaciation (2.4-2.75 Ma). The comparison between foram species living at the surface and in the (sub-)thermocline gives an estimate on how deep the thermocline was and, thus, on the intensity of

upwelling and primary productivity. The new data in this study are focused on geochemistry of the deep dwelling G. crassaformis and abundance of thermocline species. These show a shift around 2.6 Ma when the intermediate dwellers decrease in abundance and the deep-dwelling crassaformis increases suggesting a switch towards a shallower thermocline afterwards. This may have played a role in the development of larger ice sheets in the Northern Hemisphere. Interestingly, the temperature and d18O records do not show this shift but neither a clear G-IG cyclicity. A longer-term trend seems to be present, although it is unclear what could have caused this.

In general, this manuscript is well-written, concise and easy to read providing new high-resolution data for an area and time interval during which a lot was happening. What I am missing a bit is the interaction with other studies dealing with this theme to come to a clear mechanism what caused what, i.e. a shallow thermocline led to more ice build-up or the other way around? A lot of work has been done already, both specifically for this time interval, but also for the full Pliocene/Pleistocene showing how long-term trends did develop. And although several studies are mentioned throughout the text as "supporting" the new data, I think the manuscript would improve if the current study is projected for example onto the longer trends and/or compared with model experiments. This would also help to determine if an apparent trend in Dd18O and DMg/Ca fits in with the overall trend, i.e. the thermocline shoaling throughout the east Pacific in Steph et al. (2010) does show a lot of variability occurring around 2.5-2.7 Ma. So could it be that the reconstructions here are more temporal variability than a long-term change?

Additionally, the impact of the final stages of closing the Panamanian Gateway could still have been involved, both concerning productivity (Schneider and Schmittner, 2006) and thermocline structure; Groeneveld et al. (2014) also show thermocline vs surface temperatures for MIS 96-100 for east Pacific Site 1241. As Site 1241 is located outside the cold tongue, modern conditions show a strong sea water temperature gradient between both locations, but the long-term thermocline shoaling during the Pliocene occurred both in and outside the cold tongue area. A comparison would therefore also

provide additional evidence for a change in intensity of upwelling/cold tongue or if the full east Pacific experienced these changes.

Contamination of the samples seems to be absent concerning Mg/Ca, but Mn/Ca values are relatively high. These values, however, are not uncommon in older sediments (Groeneveld et al., 2006; Schmidt et al. 2006); and for the Galapagos area Lea et al. (2005) linked higher Mn/Ca to volcanic particles. One way to check the character of the Mn is to perform reduction cleaning to see if Mn-oxyhydroxides are involved. Another possibility may be an actual bottom water signal. Mn/Ca is recently receiving increasing attention as a recorder of bottom water oxygen conditions, either in the sediment, as coatings involving $MnCO_3$ being formed onto the tests or in the foram calcite itself. If that is the case you may see glacial-interglacial variability in the Mn/Ca and it may be linked to d13C as variations in productivity would change the intensity of the oxygen minimum zone.

You distinguished between dextral and sinistral forms of crassaformis where possible. Based on a previous study there was no difference in the geochemical structure, but is the occurrence of both forms controlled by glacial-interglacial variability? Also, did you notice differences in the signature of more heavily encrusted specimens vs less-encrusted specimens or even between different morphological types which occur during this period (Rögl 1974)? Seasonality of ruber: Sediment trap studies often show a distinct seasonality in ruber fluxes when areas are affected by seasonal upwelling conditions. So this may mean that also in the cold tongue ruber is more inclined towards the season when upwelling decreases in intensity (Mohtadi et al., 2009; Jonkers and Kucera, 2015).

Add bars in the figures to better be able to distinguish between glacial and interglacial time periods.

Page 8, line 23: although in the case here with ruber and crassaformis living in different water masses, your d18O may also indicate a difference in salinity.
In conclusion, I think that this manuscript is fitting very well with Climate of the Past, but the discussion could use more attention by including and comparing with existence literature. Therefore, I recommend moderate revisions.

---

## Referee Comment (RC2) · Anonymous Referee #2 · 1 Feb 2018

The paper by Jakob et al focusses on a new high resolution paleoceanographic record of the onset of the large Northern Hemisphere Glaciaitions at the Plio-Pleistocene transition 2.6 Myrs ago. The authors have documented changes in the surface hydrography at site ODP849, in the Eastern Equatorial Pacific, based on coupled d18O and Mg/Ca in G. ruber, record mostly published in previous papers by the same group, and compare this record with a new G. crassaformis d18O/Mg/Ca record interpreted as a deep thermocline species. Those geochemical datasets are augmented with a record of sediment fluxes and with some countings of G.crassaformis and G. menardii/G. tu-

mida. Using the difference in the temperature records and d18O, the authors describe what they think changes in the Eastern equatorial Pacific thermocline, with a thermocline shoaling until 2.55 Myr followed by a stable thermocline. The article is a welcome addition as it does document in the EEP a Mg/Ca record for a deep dwelling species.

The manuscript is well written and the figures are also generally well crafted. As this is the third manuscript on the same record, the paper also details what are the novelties compared to the previous records. I do feel that the technical issues are well thought-out, e.g. the potential impact of Mn crusts on the Mg composition of the foraminiferal calcite is ruled out with some backed up arguments.(but missing the Pena et al, 2005 study which worked in the EEP to estimate the impact of these crusts on the Mg/Ca of foraminifera). On the choice of the calibration used, the authors are also quite careful, and do pick the Cleroux et al calibration quite sensibly.

My main comment on the manuscript, is that it does miss a real discussion. Symptomatically, the authors did not compare their records to any other records either from the same region or from more remote sites, which would have lent some weight to their hypothesis. Their G. crassaformis record is interesting and should be more carefully addressed. I will detail a series of questions that should be addressed, from my point of view through some significant revisions :

(1) The Mg/Ca values measured in G. crassaformis are quite low, and give some very low temperature range, mostly between 1 to 6°C (regardless of the calibration used is the one by Cléroux or the one by Regenberg). Those temperatures appear to be even colder than modern temperature at the sites, and it is unlikely that the LGM temperatures were much colder than 1°C. I am thus puzzled by those extremely low temperatures, though one might argue that they are close to the Tcrassa inferred at site DSDP214. Moreover the temperatures at the site 849 are much colder than surface subantarctic waters during the same time interval (site 1090). I would like to have some sense of the process by which the water masses where crassaformis do live would be much colder in the equatorial Pacific than in subantarctic waters.

[Figure]

(2) The location of the site ODP849 is at the edge of the cold tongue. Deglacial studies have shown that this cold tongue did migrate both longitudinally, but also latitudinally (e.g. Koutavas et al. 2003). I wonder if one might not interpret the subtle changes in the record as a long term shift the EEP rather than a subsurface process.

(3) I am puzzled by the number of G. crassaformis found in the record, reaching at sometimes close to 30% of the >250$\mu$m. Though the comparison with modern and LGM census of planktonic foraminifera is not straightforward, as late Pleistocene counts are based on the >150$\mu$m fraction, I am surprised that coe tops data show extremely low percentages of G.crassaformis (typically below 1%, exceptionally reaching 5%), far less with results from this study. I understand that the authors do have some arguments that the dissolution is limited at this site (fragmentation index for example), yet I cannot find an alternate process that would selectively get rid of most of the surface to subsurface species.

(4) The $\Delta$T record does not show any glacial/interglacial dynamics. This is quite surprising as there are a large number of studies (modelling and observational) that have shown some changes in the thermocline depth during the most recent glacial/interglacial transitions. I wonder then if the choice of picking a quite deep species (see below) and a shallow species such as G. ruber does really reflect changes in the thermocline. Species such as G. tumida, G. menardii, or N. dutertrei living closer to the thermocline would have been more sensititive to changes. I would therefore be grateful if the authors could add some lines on how they can groundcheck their proxy of the thermocline ?

(5) The living depth of G. crassaformis in this study is supposed to be within the 500 to 1000 m range. To set the record straight, the authors have to be be clear that they think that the "calcification range" of G. crassaformis is within this range. All the studies quoted by the paper to posit this range come from surface sediment samples, in which the authors have made the assumption that the isotopic temperature reflects the calcification depth. This is different from the actual mean living depth. As a couple

of examples, the paper by Jones (1967) in the equatorial Atlantic did find most G. crassaformis at depths ranging 200 to 300 meters, not below 500 meters. The authors also quote Wejnert et al 2013 indicating a calcification depth below 500 meters. This is not what the paper states, as they indicate that the range is above 300 meters. Please correct accordingly.

Details :

note [page 2]: One might also consider the last major tipping climate history : the Holocene to Anthropocene transition or the last deglaciation. Please reword more carefully.

note [page 2]: I would tend to think that it is not the shallow depth of the thermocline that exerts a role in the ENSO, but rather the reverse. So please reword in thinking the Eastern Pacific Ocean as a part of the ocean where atmospheres and surface oceanic layers are subtlety interconnected.

note [page 2]: I understand the framing in two alternates hypotheses, but there is also a mid-ground solution where the state of the equatorial Pacific did play a substantial role, without being the main climatic ruler. Moreover, if one would really test the role of the EEP, he would have to reconstruct the dynamics of the equator to pole gradient.

note [page 2]: "We use planktic (both sea-surface- and thermocline-dwelling) foraminiferal geochemical ($\delta$ 18 O, $\delta$ 13 C and Mg/Ca) proxy records in combination with sedimentological (sand-accumulation rates) and faunal (abundance data of thermocline-dwelling foraminiferal species) information to reconstruct thermocline depth for the final phase of the late Pliocene/early Pleistocene iNHG from ∼2.75 to 2.4 Ma (MIS G7-95)" : This final sentence of the introduction, which sums up the methods should be either moved in the methods, or argumented.

note [page 6]: The use of this very large size fraction is not regularly used. Could you elaborate on this choice?

note [page 7]: A low fragmentation index might also correspond to the selective preservation of only resistant species. Please rephrase this sentence.

note [page 7]: Please be more specifics: what is the seasonality at the location of the site? - even though it might be significantly different, it cannot be ruled out without testing it

note [page 8]: What is the mean Temperature at the site?

note [table1 page 17]: Add the number of samples processed for each site and study to give a sense of the effort included in this study.

note [Figure 1 page 18, panel B]: A latitudinal transect would be more useful to test whether the front did change as in Koutavas et al.

---

## Author Comment (AC1) · 28 Feb 2018

**Thermocline state change in the Eastern Equatorial Pacific during the late Pliocene/early Pleistocene intensification of Northern Hemisphere Glaciation**

- Response to Reviewer #1 -

We thank Reviewer #1 for his/her careful and thorough assessment of our manuscript. Below, we provide a point-by-point response to all comments and suggestions made by Reviewer #1.

**Response to general comments**

**R.1.1:** "Thermocline state change in the Eastern Equatorial Pacific during the late Pliocene/early Pleistocene intensification of Northern Hemisphere Glaciation by Jakob et al. This manuscript uses a combination of foraminiferal geochemistry (Mg/Ca and stable oxygen and carbon isotopes), abundance and sand accumulation rate to reconstruct how the thermocline in the east Pacific cold tongue area developed during the onset of Northern Hemisphere Glaciation (2.4–2.75 Ma). The comparison between foram species living at the surface and in the (sub)-thermocline gives an estimate on how deep the thermocline was and, thus, on the intensity of upwelling and primary productivity. The new data in this study are focused on geochemistry of the deep dwelling *G. crassaformis* and abundance of thermocline species. These show a shift around 2.6 Ma when the intermediate dwellers decrease in abundance and the deep-dwelling *G. crassaformis* increases suggesting a switch towards a shallower thermocline afterwards. This may have played a role in the development of larger ice sheets in the Northern Hemisphere. Interestingly, the temperature and $\delta^{18}O$ records do not show this shift but neither a clear glacial-interglacial cyclicity. A longer-term trend seems to be present, although it is unclear what could have caused this.

In general, this manuscript is well-written, concise and easy to read providing new high-resolution data for an area and time interval during which a lot was happening. What I am missing a bit is the interaction with other studies dealing with this theme to come to a clear mechanism what caused what, i.e. a shallow thermocline led to more ice build-up or the other way around?"

We thank the reviewer for this positive assessment. Substantial shoaling of the thermocline as documented by our data from ~2.64 to 2.55 Ma suggests that major changes in thermocline depth occurred shortly before the final phase of the late Pliocene/early Pleistocene intensification of Northern Hemisphere Glaciation (iNHG) and that (sub-)tropical thermocline shoaling,

perhaps, was a precondition to allow the development of large ice sheets in the Northern Hemisphere. Our observation is in line with studies from Fedorov et al. (2006) and Dekens et al. (2007), but contradicts the results of Wara et al. (2005), Steph et al. (2010) and Ford et al. (2012). The latter have documented most prominent changes in thermocline depth only prior to ~3.5 Ma (see Fig. S2.1 in our Response Letter to Reviewer #2).

A brief discussion on what caused what, i.e., whether thermocline shoaling favoured Northern Hemisphere ice build-up or *vice versa*, as asked by the reviewer, has already been included in the original version of the manuscript (Section 5.4.2, p. 12, lines 6–10). To account for the reviewer's comment, however, we will elaborate this discussion in more detail, also in comparison with other studies as also suggested by Reviewer #2 (for details see our response to comment R.2.2 by Reviewer #2).

**R.1.2:** "A lot of work has been done already, both specifically for this time interval, but also for the full Pliocene/Pleistocene showing how long-term trends did develop. And although several studies are mentioned throughout the text as "supporting" the new data, I think the manuscript would improve if the current study is projected for example onto the longer trends and/or compared with model experiments. This would also help to determine if an apparent trend in $\Delta\delta^{18}$O and $\Delta$Mg/Ca fits in with the overall trend, i.e. the thermocline shoaling throughout the east Pacific in Steph et al. (2010) does show a lot of variability occurring around 2.5–2.7 Ma. So could it be that the reconstructions here are more temporal variability than a long-term change?"

We highly appreciate this suggestion. Our new records ($\delta^{18}$O and Mg/Ca gradients) from Site 849 indicate a general shoaling of the thermocline across the entire target interval (~2.75–2.4 Ma), and therefore can be termed a "long-term" shoaling trend, although the most prominent thermocline shoaling is restricted to the ~2.64–2.55 Ma period (Fig. 6b). The overall shoaling trend observed at Site 849 across the Plio-Pleistocene transition matches results from previous proxy records and modelling efforts (Wara et al., 2005; Fedorov et al., 2006; Dekens et al., 2007; Steph et al., 2010; Ford et al., 2012) that consistently document a "long-term" shoaling of the thermocline in the Eastern Equatorial Pacific (EEP) and other (sub-)tropical upwelling regions throughout the Plio-Pleistocene.

Upon closer inspection, however, the geochemical (Mg/Ca and $\delta^{18}$O) records of the thermocline-dwelling species *G. tumida* from Site 1241 in the East Pacific Warm Pool reveal a short-term, i.e., "transient" thermocline deepening from ~2.7 to 2.5 Ma that is superimposed

on the overall "long-term" shoaling trend observed in this study for ~6–2 Ma (Steph et al., 2010; see Fig. S2.1 in our Response Letter to Reviewer #2)).

"Transient" thermocline deepening from ~2.7 to 2.5 Ma as shown for Site 1241 (Steph et al., 2010) contradicts our proxy records for thermocline depth at Site 849 that rather indicate a "long-term" thermocline shoaling trend for this time interval; unfortunately, other records used to infer thermocline evolution in the tropical east Pacific for this interval lack the required temporal resolution (i.e., ~12.5 kyr [Site 847; Wara et al., 2005], ~10–20 kyr [multiple east Pacific sites; Dekens et al., 2007], and ~30 kyr [multiple EEP sites; Ford et al., 2012] *versus* ~750 yr [Site 849; this study] to ~3 kyr [Site 1241; Steph et al., 2010]) to resolve the "transient" thermocline state change during the ~2.7–2.5 Ma period as observed for Site 1241, if existent. Nevertheless, we hypothesize that records from Site 1241 reflect a more local signal than Site 849 and other sites located within the equatorial "cold tongue". More-over, Site 1241 was possibly temporally affected by glacioeustatically induced openings and closures of the Central American Seaway around ~2.5 Ma (Marine Isotope Stages [MIS] 100–96): High-resolution (~1 kyr) data (surface-to-thermocline, i.e., *G. sacculifer*-to-*N. dutertrei,* Mg/Ca-based temperature gradient) from this site reveal changes in thermocline depth on glacial-interglacial timescales for MIS 100–96 (Groeneveld et al., 2014), while our records from Site 849 indicate no glacial-interglacial variability in thermocline depth for the same time interval (for details see our response to comment R.1.3 and Fig. S2.1 in our Response Letter to Reviewer #2).

To account for the reviewer's comment, a more detailed comparison of our new re-cords on thermocline evolution at Site 849 to other thermocline data in the tropical east Pa-cific will be included in Sections 5.3 ("Stable-isotope and Mg/Ca records of *G. crassaformis* and *G. ruber* at Site 849") and 5.4.1 ("Geochemical evidence"), focussing on both the "long-term" trend and also on the "transient" or glacial-interglacial variability. In this context, we also plan to modify Figure 1 by showing (in addition to the two maps already presented) a global map indicating the location of sites that will be mentioned in the text. This will further enhance clarity for the readers.

**R.1.3:** "Additionally, the impact of the final stages of closing the Panamanian Gateway could still have been involved, both concerning productivity (Schneider and Schmittner, 2006) and thermocline structure; Groeneveld et al. (2014) also show thermocline *vs* surface temperatures for MIS 96–100 for east Pacific Site 1241. As Site 1241 is located outside the cold tongue, modern conditions show a strong sea water temperature gradient between both locations, but

the long-term thermocline shoaling during the Pliocene occurred both in and outside the cold tongue area. A comparison would therefore also provide additional evidence for a change in intensity of upwelling/cold tongue or if the full east Pacific experienced these changes."

We thank the reviewer for this comment. The records of Groeneveld et al. (2014) suggest that water-column stratification (defined as the temperature difference between the surface-dwelling species *G. sacculifer* and the thermocline-dwelling species *N. dutertrei*) at east Pacific Site 1241 was higher during interglacials (temperature difference [$\Delta$T] of ~3–4 °C) than during glacials ($\Delta$T of ~1–2 °C) for MIS 100–96, thereby implying thermocline-state changes on glacial-interglacial timescales (see Fig. S2.1 in our Response Letter to Reviewer #2). This contradicts our new data from Site 849 where both surface-to-thermocline $\delta^{18}$O and temperature gradients indicate no change in thermocline depth on glacial-interglacial timescales for MIS G6–96 (Fig. 6b). Together, data from Sites 849 and 1241 suggest that thermocline-state changes have not occurred homogenously across the entire tropical east Pacific, possibly due to the following reasons:

(i) Sites 849 and 1241 are located in different areas of the highly heterogeneous tropical east Pacific (i.e., cold tongue upwelling region [Site 849] *versus* warm pool [Site 1241]). Therefore it appears reasonable to hypothesize that changes in thermocline depth in- and outside the cold tongue upwelling regime simply underwent a different evolution on glacial-interglacial timescales. This hypothesis is supported by faunal data (calcareous nan-nofossil counts) from EEP cold tongue Site 846, which, like Site 849, do not indicate any variations in thermocline depth for MIS 101–95 (Bolton et al., 2010).

(ii) It is likely that Site 1241 reflects a more local signal due to influences by glacioeustatical-ly induced openings and closures of the Central American Seaway during MIS 100–96 (as suggested by Groeneveld et al. [2014]), while Site 849 records a more open-ocean, quasi-global signal  due to its position west of the East Pacific Rise (Mix et al., 1995). This is, for example, highlighted by the fact that global climate-ocean ecosystem model experiments (Schneider and Schmittner, 2006) indicate that a closure of the Central American Seaway helps to promote upwelling in nearby EEP regions (including Site 1241) – in contrast, Site 849 appears to be mostly unaffected by such changes as shown by sand-accumulation-rate-based primary productivity data (Jakob et al., 2016). However, it is likely, although specu-lative, that temporal closures of the Central American Seaway from MIS 100 through 96 (Groeneveld et al., 2014) might have contributed to generally increasing upwelling and thus primary productivity rates that have been observed at Site 849 since MIS 100 (Jakob et al., 2016; this study; Fig. 6c–d).

To account for this comment, we will include a discussion into Sections 5.4.1 ("Geochemical evidence") and 5.4.2 ("Faunal and sedimentological evidence") on (i) thermocline data derived from Sites 849 and 1241 (also in comparison to other datasets from the east Pacific; for details see our response to comment R.1.2 by Reviewer #1 and to comments R.2.2 and R.2.3 by Reviewer #2; see also Fig. S2.1 in our Response Letter to Reviewer #2), and on (ii) the impact of glacioeustatically induced openings and closures of the Central American Seaway to these sites.

**R.1.4:** "Contamination of the samples seems to be absent concerning Mg/Ca, but Mn/Ca values are relatively high. These values, however, are not uncommon in older sediments (Groeneveld et al., 2006; Schmidt et al. 2006); and for the Galapagos area Lea et al. (2005) linked higher Mn/Ca to volcanic particles. One way to check the character of the Mn is to perform reduction cleaning to see if Mn-oxyhydroxides are involved. Another possibility may be an actual bottom water signal. Mn/Ca is recently receiving increasing attention as a recorder of bottom water oxygen conditions, either in the sediment, as coatings involving $MnCO_3$ being formed onto the tests or in the foram calcite itself. If that is the case you may see glacial interglacial variability in the Mn/Ca and it may be linked to $\delta^{13}C$ as variations in productivity would change the intensity of the oxygen minimum zone."

We appreciate this comment. As highlighted in our manuscript, it is important to note that if Mn-rich overgrowths existed on *G. crassaformis* tests, they might change absolute temperature estimates by on average ~0.5 °C, but this would not affect the overall shape of the Mg/Ca-based temperature record (Fig. 4c, d). Therefore it appears reasonable to assume that early diagenetic Mn-rich overprinting has no significant impact on our interpretation regarding relative changes in the surface-to-thermocline gradient. Hence, we argue that for the purpose of our study the reductive cleaning step, which might remove such overgrowths, is not required. However, to account for the reviewer's comment, we have re-checked the character of Mn in/on *G. crassaformis* tests at our study site as follows:

(i) The reviewer suggests that Mn-oxyhydroxides might perhaps explain enhanced Mn/Ca ratios (~0.2–2.1 mmol/mol; Fig. 4) in *G. crassaformis* tests at our study site. We have tested this possibility through a modification of the cleaning procedure for element analyses via including an additional reductive cleaning step for selected samples (e.g., Barker et al., 2003). As a reductive reagent, a mixture of hydrazine, ammonium hydroxide and ammonium citrate was used. The results show by ~40–45 % lower Mn/Ca values for samples that were cleaned reductively compared to samples that underwent only oxidative cleaning

(Fig. S1.1). This indicates that at least two different Mn phases coexist on *G. crassaformis* tests surfaces – perhaps Mn-oxyhydroxides that can be removed by reductive cleaning and another phase that can neither be removed by oxidative nor by reductive cleaning. To comply with the reviewer's comment, we will include this information into Section 5.2 ("Assessment of contamination and diagenetic effects on Mg/Ca ratios of *G. crassaformis*").

[Figure]

**Figure S1.1: Evaluation of potential contaminations on *G. crassaformis* test surfaces from Site 849.** Mn/Ca ratios of selected samples (1: 849D-7/3-77–79cm, 2: 849D-7/3-105–107cm, 3: 849D-7/4-51–53cm, 4: 849D-7/4-67–69cm) that underwent reductive and oxidative cleaning (red) or oxidative cleaning only (blue).

(ii) The reviewer is correct in stating that if foraminiferal Mn/Ca ratios trace bottom-water oxygen conditions (e.g., McKay et al., 2015; Koho et al., 2017), our Mn/Ca record should follow the glacial-interglacial cyclicity given by the productivity proxy record from the same site and time interval (Fig. 6d; Jakob et al., 2016). To identify coherencies between these two records (Mn/Ca values *versus* sand-accumulation rates; note that we use the latter record as a primary productivity proxy instead of commonly used foraminiferal $\delta^{13}$C proxy data since $\delta^{13}$C at Site 849 has been shown to not simply trace *in-situ* changes in productivity, but rather reflects the $\delta^{13}$C signature imprinted on high-southern-latitude waters that are transported to the EEP [Jakob et al., 2016]) we performed Blackman-Tukey cross-spectral analyses with a 30 % overlap using the AnalySeries software package version 2.0.8 (Paillard et al., 1996). Data for cross-spectral analyses were linearly interpolated, detrended, and prewhitened.

Our results indicate that indeed both records yield a 41-kyr (i.e., glacial-interglacial) cyclicity (Fig. S1.2). However, the Mn/Ca and the productivity proxy records do not fluctuate in phase; instead, they are shifted by approximately -102° (equal to -11.5 kyr) for the 41 kyr period. If Mn/Ca reflects bottom-water oxygen concentrations, the oxygen

(Mn/Ca) and productivity (sand-accumulation rate) proxy records are, however, expected to fluctuate in phase. Therefore we exclude that *G. crassaformis* Mn/Ca data at Site 849 record bottom-water oxygenation. We decided not to include this discussion into the manuscript as it goes far beyond the scope of our study, and we hope this finds approval of the editor.

[Figure]

**Figure S1.2. Blackman-Tukey cross-spectral analysis for the identification of phase shifts at Site 849 for the time interval from ~2.65 to 2.4 Ma.** Coherence (green) and phase (purple) relationship between productivity (sand-accumulation rates [Jakob et al., 2016]) and Mn/Ca ratios are plotted on log scales. Negative values in this phase plot indicate that productivity lags Mn/Ca, i.e., by -102° (equal to -11.5 kyr) for the 41 kyr period.

(iii) Reviewer #1 suggests that enhanced Mn/Ca values in our samples could also be linked to volcanic particles (e.g., Lea et al., 2005). However, while carefully reading this study, we recognized that Lea et al. (2005) observed enhanced Fe/Ca and Al/Ca ratios (rather than increased Mn/Ca ratios) in planktic foraminifera from core intervals that were rich in volcanic debris. In contrast, they link higher Mn/Ca values to diagenetic coatings (Mn-carbonates) on foraminiferal tests. Moreover, we are unaware of any study that has shown a relation between foraminiferal Mn/Ca ratios and volcanic material. Therefore we decide to not include such a discussion into our manuscript.

**R.1.5:** "You distinguished between dextral and sinistral forms of *G. crassaformis* where possible. Based on a previous study there was no difference in the geochemical structure, but is the occurrence of both forms controlled by glacial-interglacial variability? Also, did you notice differences in the signature of more heavily encrusted specimens *vs* less encrusted specimens or even between different morphological types which occur during this period (Rögl 1974)?”

We preferentially selected the most abundant coiling direction (sinistral) of *G. crassaformis* tests at Site 849. However, the sinistral coiling forms did not occur continuously over the sampling interval. Thus, we also used the dextral coiling tests for some intervals, (~2.46–2.43 Ma [69.84–69.04 meters composite depth (mcd)], ~2.51–2.50 Ma [70.85–70.68 mcd]) and also for some specific time points (2.40 Ma [68.10 mcd], 2.41 Ma [68.32 mcd]). The occurrence of dextrally and sinistrally coiling *G. crassaformis* specimens, however, appears to be randomly rather than following a certain cyclicity.

Furthermore, we did not notice a change between heavily and less encrusted *G. crassaformis* specimens or even between different morphotypes throughout the investigated time interval while picking. Scanning Electron Microscope (SEM) images of *G. crassaformis* specimens derived from both glacial and interglacial samples also do not reveal such changes (Fig. 3).

To account for the reviewer's comment and to enhance clarity for the readers, we will modify the relevant sentences in Sections 4.1 (“Sample material”) and 5.1 (“Foraminiferal test preservation at Site 849”) when revising our manuscript.

**R.1.6:** “Seasonality of *G. ruber*: Sediment trap studies often show a distinct seasonality in *G. ruber* fluxes when areas are affected by seasonal upwelling conditions. So this may mean that also in the cold tongue *G. ruber* is more inclined towards the season when upwelling decreases in intensity (Mohtadi et al., 2009; Jonkers and Kucera, 2015).”

In extratropical regions *G. ruber* most likely represents summer surface-water conditions (e.g., Schiebel and Hemleben, 2000; Schiebel et al., 2002), while in the tropics, where seasonal climate variability is low, plankton tow studies indicate that this species can be found year around and is therefore typically used as recorder for mean annual surface-water conditions (e.g., Deuser et al., 1981; Lin et al., 1997). This is supported by a study on global planktic foraminiferal shell fluxes that show that in waters with temperatures >25 °C (i.e., sea-surface temperatures [SSTs] prevailing at our study site during interglacials; Jakob et al. [2017]) *G. ruber* has a less predictable flux pattern with random peak timing (Jonkers and Kučera, 2015). In colder waters, seasonality appears to be more prominent, with decreasing fluxes when temperature decreases (Jonkers and Kučera, 2015). The reviewer is therefore correct in stating that *G. ruber* proxy data (particularly of glacial periods) might be biased towards the season with lower upwelling intensity. In contrast, however, Mohtadi et al. (2009)

show that *G. ruber* fluxes in the upwelling region off south Java are not solely related to up-welling: Although this species consistently shows enhanced flux rates during the upwelling period, its flux rates are also temporarily as high during non-upwelling seasons.

In light of the above considerations, we find it reasonable to assume that *G. ruber* is a more likely recorder for mean surface-water conditions than for a particular season in the low-latitude EEP upwelling regime. However, to account for the reviewer's comment, relevant references will be added to Section 5.3 ("Stable-isotope and Mg/Ca records of *G. crassaformis* and *G. ruber* at Site 849"), and more detailed information on seasonality at Site 849 will be provided (for details see our response to comment R.2.14 by Reviewer #2).

**R.1.7:** "Add bars in the figures to better be able to distinguish between glacial and interglacial time periods."
Bars as suggested by the reviewer had already been included in the figures of the original manuscript version. However, we will act on this suggestion by highlighting the bars by a more prominent colour.

**Response to line by line remarks:**

**R.1.8:** "Page 8, line 23: although in the case here with *G. ruber* and *G. crassaformis* living in different water masses, your $\delta^{18}$O may also indicate a difference in salinity."
We fully agree with the reviewer and will revise this paragraph.

**References (other than those already cited in the discussion paper)**

Deuser, W. G., Ross, E. H., Hemleben, C., and Spindler, M.: Seasonal changes in species composition, numbers, mass, size, and isotopic composition of planktonic foraminifera settling into the deep Sargasso Sea, Palaeogeogr., Palaeoclimat., Palaeoecol., 33, 103–127, 1981.

Koho, K. A., de Nooijer, L. J., Fontanier, C., Toyofuku, T., Oguri, K., Kitazato, H., and Reichart, G.-J.: Benthic foraminiferal Mn/Ca ratios reflect microhabitat preferences, Biogeosciences, 14, 3067–3083, 2017.

Lea, D. W., Pak, D. K., and Paradis, G.: Influence of volcanic shards on foraminiferal Mg/Ca in a core from the Galápagos region, Geochem., Geophys., Geosyst., Q11P04, doi:10.1029/2005GC000970, 2005.

McKay, C. L., Groeneveld, J., Filipsson, H. L., Gallego-Torres, D., Whitehouse, M. J., Toyofuku, T., and Romero, O. E.: A comparison of benthic foraminiferal Mn/Ca and sedimentary Mn/Al as proxies of

relative bottom-water oxygenation in the low-latitude NE Atlantic upwelling system, Biogeosciences, 12, 5415–5428, 2015.

Rögl, F.: The evolution of the *Globorotalia truncatulinoides* and *Globorotalia crassaformis* group in the Pliocene and Pleistocene of the Timor Through, DSDP Leg 27, Site 262, Init. Repts., 27, Washington, 1974.

Schiebel, R., and Hemleben, C.: Modern planktic foraminifera. Paläontologische Zeitschrift, 79, 135–148, 2005.

Schiebel, R., Schmuker, B., Alves, M., and Hemleben, C.: Tracking the recent and late Pleistocene Azores front by the distribution of planktic foraminifers. J. Marine Syst., 37, 213–227, 2002.

---

## Author Comment (AC2) · 28 Feb 2018

**Thermocline state change in the Eastern Equatorial Pacific during the late Pliocene/early Pleistocene intensification of Northern Hemisphere Glaciation**

- Response to Reviewer #2 -

We thank Reviewer #2 for his/her careful and thorough assessment of our manuscript. Below, we provide a point-by-point response to all comments and suggestions made by Reviewer #2.

**Response to general comments**

**R.2.1:** "The paper by Jakob et al focusses on a new high resolution paleoceanographic record of the onset of the large Northern Hemisphere Glaciations at the Plio-Pleistocene transition 2.6 Myrs ago. The authors have documented changes in the surface hydrography at Site 849, in the Eastern Equatorial Pacific, based on coupled $\delta^{18}$O and Mg/Ca in *G. ruber*, record mostly published in previous papers by the same group, and compare this record with a new *G. crassaformis* $\delta^{18}$O/Mg/Ca record interpreted as a deep-thermocline species. Those geochemical datasets are augmented with a record of sediment fluxes and with some countings of *G. crassaformis* and *G. menardii/G. tumida*. Using the difference in the temperature records and $\delta^{18}$O, the authors describe what they think changes in the EEP thermocline, with a thermocline shoaling until 2.55 Myr followed by a stable thermocline. The article is a welcome addition as it does document in the EEP a Mg/Ca record for a deep-dwelling species.

The manuscript is well written and the figures are also generally well crafted. As this is the third manuscript on the same record, the paper also details what are the novelties compared to the previous records. I do feel that the technical issues are well thought out, e.g. the potential impact of Mn crusts on the Mg composition of the foraminiferal calcite is ruled out with some backed up arguments (but missing the Pena et al., 2005 study which worked in the EEP to estimate the impact of these crusts on the Mg/Ca of foraminifera). On the choice of the calibration used, the authors are also quite careful, and do pick the Cléroux et al. calibration quite sensibly."

We thank the reviewer for this positive assessment.

**R.2.2:** "My main comment on the manuscript, is that it does miss a real discussion. Symptomatically, the authors did not compare their records to any other records either from the same region or from more remote sites, which would have lent some weight to their hypothesis."

We agree with Reviewer #2 and acknowledge that our original manuscript has not presented a comparison of thermocline data from our study site with other datasets revealing thermocline evolution for the same time interval. To account for this comment made by both reviewers, we will compare thermocline data from Site 849 to the following datasets of the same time interval (to provide maximum clarity for the readers, we plan to also modify Figure 1 by showing – in addition to the two maps already presented – a global map indicating the location of all sites that will be mentioned in the text):

(i)   Geochemical ($\delta^{18}$O and Mg/Ca) data of *G. crassaformis* from eastern tropical Indian Ocean Site 214 (Karas et al., 2009; Fig. S2.1c). For a detailed discussion see our response to comment R.2.3.

(ii)  Geochemical ($\delta^{18}$O and Mg/Ca) data of *G. tumida* from eastern tropical Pacific Site 1241 (Steph et al., 2010; Fig. S2.1c). For a detailed discussion see our response to comment R.1.2 by Reviewer #1.

(iii) Surface-to-thermocline (*G. sacculifer* to *N. dutertrei*) Mg/Ca-based temperature gradients from eastern tropical Pacific Site 1241 (Groeneveld et al., 2014; Fig. S2.1a,b). For a detailed discussion see our response to comment R.1.3 by Reviewer #1.

(iv)  Alkenone-based sea-surface temperatures (SSTs) from Site 1090 in the Southern Ocean (i.e., the source region for waters upwelled in the Eastern Equatorial Pacific [EEP]) (Martínez-Garcia et al., 2010). For a detailed discussion see our response to comment R.2.3.

We note that in comparison to the Mg/Ca-based temperature data of *G. tumida* yet available for our study site and target interval (~15.5–17.5 °C; Ford et al. [2012]; Fig. S2.1c) our *G. crassaformis* temperatures (1.0–11.6 °C) appear to be relatively low. The *G. tumida*-based temperature record of Ford et al. (2012) is, however, of low temporal resolution, with only six datapoints for the 2.75–2.4 Ma interval. Therefore we argue that this comparison is not robust enough to warrant further discussion in the revised version of the manuscript.

**R.2.3:** "The Mg/Ca values measured in *G. crassaformis* are quite low, and give some very low temperature range, mostly between 1 to 6 °C (regardless of the calibration used is the one by Cléroux or the one by Regenberg). Those temperatures appear to be even colder than modern temperature at the sites, and it is unlikely that the LGM temperatures were much colder than 1. I am thus puzzled by those extremely low temperatures, though one might argue that they are close to the Tcrassa inferred at site DSDP214. Moreover the temperatures at the site 849 are much colder than surface subantarctic waters during the same time interval (site 1090). I would like to have some sense of the process by which the water masses where

[Figure]

**Figure S2.1: Comparison of thermocline proxy records**. (a) Stratification data (surface-to-thermocline temperature gradient) for Site 849 in the EEP upwelling regime (this study; red) and Site 1241 in the East Pacific Warm Pool (Groeneveld et al., 2014; brown) for ~2.7–2.4 Ma. (b) Thermocline temperatures based on *G. crassaformis* at Site 849 in the EEP upwelling regime (this study; purple) and based on *N. dutertrei* at Site 1241 in the East Pacific Warm Pool (Groeneveld et al., 2014; orange) for ~2.7–2.4 Ma (c) Thermocline temperatures from Site 214 in the tropical eastern Indian Ocean (based on *G. crassaformis*; Karas et al., 2009; blue), Site 1241 in the East Pacific Warm Pool (based on *G. tumida*; Steph et al., 2010; green) and Site 849 in the EEP upwelling regime (based on *G. tumida*; Ford et al., 2012; black) for the past ~6 Myr. Blue bars in (a) and (b) highlight glacial periods. Yellow bar in (c) marks our study interval.

*G. crassaformis* do live would be much colder in the equatorial Pacific than in subantarctic waters."

We agree with Reviewer #2 that temperatures reconstructed from *G. crassaformis* (∼1–11.6 °C; Fig. 5d) are difficult to reconcile with

(i) thermocline temperatures of the Last Glacial Maximum in the eastern tropical Pacific (∼14 to 16 °C inferred from Mg/Ca data of *N. dutertrei* from the Cocos and Carnegie Ridges [Hertzberg et al., 2016] and of *G. tumida* from Site 849 [Ford et al., 2015], respectively);

(ii) modern bottom-water temperatures at our study site of about ∼1.5 °C (Locarnini et al., 2013) – note that because the exact calcification depth of *G. crassaformis* in the EEP remains unclear, a direct comparison to modern temperatures of an assumed *G. crassaformis* calcification depth (~400–800 m, equal to ~5–8 °C at Site 849; Fig. 1b) is not straight forward (for details see our response to comment R.2.15);

(iii) SSTs of ∼10.5–17 °C during our study interval at Site 1090 (Martínez-Garcia et al., 2010) in the Southern Ocean that provides source waters for the EEP upwelling region (we are aware that Site 1090 is not on the direct trajectory from the Southern Ocean to the EEP; however, this site is accepted as the best end member of Southern Ocean waters currently available [Billups et al., 2002; Pusz et al., 2011]).

Although our temperature record of *G. crassaformis* appears to be too cold in comparison to the above-mentioned data, it is close to the temperature range inferred for Site 214 in the tropical eastern Indian Ocean for the same species and time interval (∼8–10 °C; Karas et al, 2009; Fig. S2.1). This makes it reasonable to assume that our *G. crassaformis*-based temperature record indeed reflects realistic values. However, thermocline temperatures of ∼1–11.6 °C at low-latitude Sites 214 and 849 are difficult to reconcile with SSTs of ∼10.5–17 °C at high-southern-latitude Site 1090 during the same time interval. This temperature difference implies that there must have been substantial cooling of Southern Ocean surface-waters (i) either when being downwelled and transported to the lower latitudes and/or (ii) through mixing with Antarctic Bottom Waters, which presently has a bottom-water temperature of about 0 °C (Craig and Gordon, 1965). In this context, it is important to note that very low *G. crassaformis*-based temperatures at Site 849 of ∼1 °C result from only one datapoint, while temperatures typically higher than 2–3 °C during the remaining target interval even during most prominent glacials of the intensification of Northern Hemisphere Glaciation can be reconciled better with the mixing hypothesis presented above.

To account for the reviewer's comment, we will briefly elaborate on absolute temperature estimates derived from *G. crassaformis* in comparison to other datasets. This discussion will go into Section 5.3 ("Stable-isotope and Mg/Ca records of *G. crassaformis* and *G. ruber* at Site 849").

**R.2.4:** "The location of the Site ODP849 is at the edge of the cold tongue. Deglacial studies have shown that this cold tongue did migrate both longitudinally, but also latitudinally (e.g. Koutavas et al. 2003). I wonder if one might not interpret the subtle changes in the record as a long term shift the EEP rather than a subsurface process."

We agree that there is a migration of the cold tongue over glacial-interglacial timescales that can be reconstructed using an approach as exemplified in the study of Koutavas et al. (2003). For the studied time interval, however, the amount of sites (both with the required temporal resolution and preservation of foraminiferal tests) that would be required for such a study is simply not available. Furthermore, the migration mentioned by Reviewer #2 occurs on a glacial-interglacial timescale, while the changes inferred from our data show no significant changes on these short timescales, but rather a long-term shift in thermocline depth. Therefore, we acknowledge the comment of Reviewer #2, but are unfortunately not able to include this kind of information in our contribution.

**R.2.5:** "I am puzzled by the number of *G. crassaformis* found in the record, reaching at sometimes close to 30 % of the >250 µm. Though the comparison with modern and LGM census of planktonic foraminifera is not straight forward, as late Pleistocene counts are based on the >150 µm fraction, I am surprised that core-top data show extremely low percentages of *G. crassaformis* (typically below 1 %, exceptionally reaching 5 %), far less with results from this study. I understand that the authors do have some arguments that the dissolution is limited at this site (fragmentation index for example), yet I cannot find an alternate process that would selectively get rid of most of the surface to subsurface species."

We respectfully disagree with the reviewer that high percentages of *G. crassaformis* within the planktic foraminiferal assemblage observed at Site 849 can only be explained by selective dissolution of less resistant planktic foraminifera such as *Globigerinoides* (Dittert et al., 1999). The overall good preservation and lack of significant changes in preservation between glacials and interglacials (Fig. 3) along with the low fragmentation index (Jakob et al., 2017) argue against preservation playing a significant role. Another possibility might be that living conditions (e.g., oxygen content in case of *G. crassaformis* [Jones, 1967; Kemle von Mücke

and Hemleben, 1999]) changed over time. In our manuscript we argue that increasing *G. crassaformis* abundances from ~2.64 Ma (MIS G2) onward (Fig. 6c) can be explained by changing environmental conditions that were more favourable for this species. This line of argument can also be used to explain lower percentages of *G. crassaformis* in the modern as opposed to our target interval assuming less favourable living conditions.

**R.2.6:** "The ΔT record does not show any glacial/interglacial dynamics. This is quite surprising as there are a large number of studies (modelling and observational) that have shown some changes in the thermocline depth during the most recent glacial/interglacial transitions. I wonder then if the choice of picking a quite deep species (see below) and a shallow species such as *G. ruber* does really reflect changes in the thermocline. Species such as *G. tumida, G. menardii,* or *N. dutertrei* living closer to the thermocline would have been more sensitive to changes. I would therefore be grateful if the authors could add some lines on how they can groundcheck their proxy of the thermocline?"

Again, we thank the reviewer for this constructive comment. First, we want to shed light onto the applicability of our approach for surface-to-thermocline (i.e., *G. ruber* to *G. crassaformis*) gradient calculation. Subsequently, we present a more detailed discussion of our ΔT record also in comparison to previous studies:

(i)   To reconstruct thermocline changes, we decided to focus on $\delta^{18}O$ and temperature gradients between sea-surface and thermocline waters. *Globigerinoides ruber* lives in the mixed layer, and as one of the shallowest-dwelling species among modern planktic foraminifera it is typically used as recorder for surface-water (mixed-layer) conditions (e.g., Dekens et al., 2002; Rippert et al., 2016); therefore we decided for this species as surface-water recorder.

As a recorder for thermocline waters, we selected *G. crassaformis* because ongoing studies suggest that this species is rather conservative in its preferred calcification depths, although the life cycle of deep-dwelling planktic foraminifera can involve a vertical migration through the water column of several hundred meters. Instead, *G. crassaformis* specimens are suggested to mainly calcify below the thermocline (Regenberg et al., 2009, Steph et al., 2009, and references therein) by maintaining a more constant depth habitat near the base of the thermocline through time compared to other deep-dwellers (Cléroux and Lynch-Stieglitz, 2010). Therefore, it is reasonable to assume that this species is a suitable recorder for deep-thermocline conditions, which led us to compare surface-water records of *G. ruber* to records of the deep-thermocline-dwelling species *G. crassaformis*

rather than to intermediate-dwelling species such as *G. tumida*, *G. menardii*, or *N. dutertrei* as a proxy for thermocline state changes.

Moreover, the selected approach has been successfully applied to previous studies on surface-water structure and thermocline evolution (e.g., Karas et al., 2009; Bahr et al., 2011). To account for the reviewer's comment and to enhance clarity for the readers, we will include information on the overall applicability of the selected approach for tracing thermocline changes into Sections 3 ("Investigated foraminiferal species") and 5.4.1 ("Geochemical evidence").

(ii) The reviewer is correct in stating that a number of previous studies showed thermocline changes on glacial-interglacial timescales across the Last Glacial Maximum in the tropical Pacific. However, it remains unclear whether the thermocline shoaled (e.g., Andreasen and Ravelo, 1997; Lynch-Stieglitz et al., 2015) or deepened (DiNezio et al., 2011) during the Last Glacial Maximum. Across our study interval, there are both model- and proxy-based studies that indicate no glacial-interglacial changes in thermocline depth in the EEP (Lee and Poulson, 2005; Bolton et al., 2010; Jakob et al., 2017). In accordance with these studies, we therefore suggest that our new $\Delta T$ (and $\Delta\delta^{18}O$) records indeed reflect a true signal of thermocline development for the EEP cold tongue, i.e., no change in thermocline depth on the glacial-interglacial timescale during the late Pliocene to early Pleistocene (Fig. 6b). Instead, work of Groeneveld et al. (2014), which hints at glacial-interglacial changes in thermocline depth at Site 1241 (Fig. S2.1), is based on a site located outside the equatorial upwelling regime. Data from this site might reflect a more local signal than sites from the EEP cold tongue, possibly rather being related to glacioeustatically induced openings and closures of the Central American Seaway during that time (for details see our response to comment R.1.3 by Reviewer #1).

To account for the comment of Reviewer #2, we will extend the paragraph dealing with the glacial-interglacial evolution of the thermocline at Site 849 and its comparison to other datasets from the same time interval (Section 5.4.1, "Geochemical evidence").

**R.2.7:** "The living depth of *G. crassaformis* in this study is supposed to be within the 500 to 1000 m range. To set the record straight, the authors have to be clear that they think that the "calcification range" of *G. crassaformis* is within this range. All the studies quoted by the paper to posit this range come from surface sediment samples, in which the authors have made the assumption that the isotopic temperature reflects the calcification depth. This is different from the actual mean living depth. As a couple of examples, the paper by Jones (1967)

in the equatorial Atlantic did find most *G. crassaformis* at depths ranging 200 to 300 meters, not below 500 meters. The authors also quote Wejnert et al. 2013 indicating a calcification depth below 500 meters. This is not what the paper states, as they indicate that the range is above 300 meters. Please correct accordingly."

We agree with Reviewer #2 and have to acknowledge that we incorrectly mixed the terms "habitat depth" and "calcification depth"; the relevant paragraph will be corrected as follows:

(i)   In accordance with our response to comment R.2.6, we will state that *G. crassaformis* lives at the bottom of the thermocline (Niebler et al., 1999; Regenberg et al., 2009, Steph et al., 2009, and references therein) with a rather conservative calcification depth as opposed to other deep-dwelling foraminiferal species (Cléroux and Lynch-Stieglitz 2010).

(ii)  Since the exact calcification depth of *G. crassaformis* in the EEP is unclear, we will use calcification depths of this species determined by $\delta^{18}$O values in the (sub-)tropical Atlantic and Caribbean Sea (~400 to 800 m water depth) (Steph et al., 2006; Regenberg et al., 2009; Steph et al., 2009; Cléroux et al., 2013).

(iii) Finally, we will also note that there are other parts in the ocean where the calcification depth of *G. crassaformis* seems to be shallower (<300 m), such as in the Cariaco Basin (Tedesco et al., 2007; Wejnert et al., 2013).

**Response to line by line remarks:**

**R.2.8:** "note [page 2]: One might also consider the last major tipping climate history: The Holocene to Anthropocene transition or the last deglaciation. Please reword more carefully."
We agree with Reviewer #2 and will reword the relevant sentences as suggested.

**R.2.9:** "note [page 2]: I would tend to think that it is not the shallow depth of the thermocline that exerts a role in the ENSO, but rather the reverse. So please reword in thinking the Eastern Pacific Ocean as a part of the ocean where atmospheres and surface oceanic layers are subtlety interconnected."
Again, we agree with the reviewer and will rephrase the respective paragraphs in the abstract and introduction as suggested.

**R.2.10:** "note [page 2]: I understand the framing in two alternate hypotheses, but there is also a mid-ground solution where the state of the equatorial Pacific did play a substantial role,

without being the main climatic ruler. Moreover, if one would really test the role of the EEP, he would have to reconstruct the dynamics of the equator to pole gradient."

Indeed, such a test of the role of the EEP would require a much larger dataset and comparison to other sites and records. Since this would be beyond the scope of our contribution, we will rephrase the respective paragraph to account for this comment by Reviewer #2.

**R.2.11:** "note [page 2]: 'We use planktic (both sea-surface- and thermocline-dwelling) foraminiferal geochemical ($^{18}$O, $^{13}$C and Mg/Ca) proxy records in combination with sedimentlogical (sand-accumulation rates) and faunal (abundance data of thermocline-dwelling foraminiferal species) information to reconstruct thermocline depth for the final phase of the late Pliocene/early Pleistocene iNHG from 2.75 to 2.4 Ma (MIS G7–95)': This final sentence of the introduction, which sums up the methods should be either moved in the methods, or argumented."

In line with the reviewer's comment, this paragraph will be moved to Section 4 ("Material and methods").

**R.2.12:** "note [page 6]: The use of this very large size fraction is not regularly used. Could you elaborate on this choice?"

We are admittedly not sure if Reviewer #2 refers to the size fraction of *G. crassaformis* tests (315–400 µm) used for geochemical analyses or the size fraction selected for foraminiferal abundance counts (>250 µm). Therefore we will briefly elaborate on both aspects in the following:

(i)  Geochemical analyses of *G. crassaformis* tests (315–400 µm):

A study that explores the relationship between the shell size of *G. crassaformis* and $\delta^{18}$O, $\delta^{13}$C and Mg/Ca shows less variations for all parameters for the >300 µm fraction as opposed to the <300 µm fraction (Elderfield et al., 2002) and validates the use of a size fraction >300 µm. Finally, the 315–400 µm fraction has been selected in order to keep ontogenetic effects as small as possible (Elderfield et al., 2002; Friedrich et al., 2012), but at the same time to allow a sufficient number of *G. crassaformis* tests per sample. Moreover, the selected size fraction has also typically been used for geochemical analyses of *G. crassaformis* tests in earlier studies (e.g., Steph et al., 2006; Karas et al., 2011).

It is further important to note that the geochemical records of *G. crassaformis* presented in Jakob et al. (2016) to be complemented in this study are based on the 315–400 µm

size fraction as well. To remain consistent, we used the same size fraction in this study. To provide maximum clarity for the readers, we will rephrase the relevant sentences.

(ii) Foraminiferal abundance counts (>250 µm):

There are studies in which the >250 µm fraction is used for abundance counts of thermocline-dwelling species (e.g., Sexton and Norris, 2008). However, the reviewer is correct in stating that for abundance counts of the entire planktic foraminiferal assemblage (i.e., thermocline- and surface-dwelling species) smaller fractions (>63 µm or >125 µm) are typically investigated (e.g., Jehle et al., 2015; Luciani et al., 2017). The use of different size fractions for the purpose of abundance counts on thermocline-dwelling *versus* all planktic species is justified by the fact that test sizes are typically larger (smaller) in thermocline-dwelling (surface-dwelling) species (e.g., Davis, et al., 2013, and references therein; Feldmeijer et al., 2015). To account for the reviewer's comment, relevant references will be added.

**R.2.13:** "note [page 7]: A low fragmentation index might also correspond to the selective preservation of only resistant species. Please rephrase this sentence."

The reviewer is correct in stating that a low planktic foraminiferal fragmentation index might indicate either a good preservation of the entire planktic foraminiferal assemblage or the selective preservation of only resistant species, while less resistant species have been dissolved. However, we suggest that a low planktic foraminiferal fragmentation index at Site 849 indicates a generally good foraminiferal preservation, because also less resistant species such as *Globigerinoides* (see for example Tab. 1 in Dittert et al., 1999) occur in large numbers in these samples and are also well preserved (Fig. 3). Instead, if a low planktic foraminiferal fragmentation index indicated selective preservation of only resistant species, a substantially reduced number of *Globigerinoides* (and a poor preservation state of those individuals preserved) would be expected.

To account for the reviewer's comment, the relevant sentence in Section 5.1 ("Foraminiferal test preservation at Site 849") will be rephrased as suggested.

**R.2.14:** "note [page 7]: Please be more specific: What is the seasonality at the location of the site? Even though it might be significantly different, it cannot be ruled out without testing it."

Modern seasonal changes in SSTs at Site 849 have an amplitude of about ~0.4 °C (~24 °C during summer [June] *versus* ~23.6 °C during winter [January]; Locarnini et al. [2013]). The same seasonality (amplitude of ~0.4 °C) has been observed for 20 m water depth (Locarnini

et al., 2013), i.e, the assumed mean depth habitat of the herein investigated foraminiferal species *G. ruber* (Wang, 2000). Determining seasonal temperature variations at a depth corresponding to the calcification depth of *G. crassaformis* is not straight forward since its calcification depth remains uncertain in the EEP (for details see our response to comment R.2.15). We suggest, however, seasonality to decrease with increasing water depth and therefore to be less than ∼0.4 °C.

The above-mentioned data indicate that seasonal temperature variability is relatively low at our study site; therefore the geochemical signatures of the investigated foraminiferal species are considered not to be seasonally biased (Lin et al., 1997; Tedesco et al., 2007; Mohtadi et al., 2009; Jonkers and Kučera, 2015). As suggested by the reviewer, more specific information on seasonality at our study site as described above will be added to Section 5.3 ("Stable-isotope and Mg/Ca records of *G. crassaformis* and *G. ruber* at Site 849").

**R.2.15:** "note [page 8]: What is the mean temperature at the site?"

Modern mean annual SSTs at Site 849 are ∼23.5 °C (Locarnini et al., 2013) at 20 m water depth, i.e., the assumed mean depth habitat of *G. ruber* (Wang, 2000) (see also Fig. 1b). Mean SSTs reflected by *G. ruber* (∼24 °C [Jakob et al., 2017]) indicate slightly warmer values, supporting the overall notion of a warmer-than-present EEP during the late Pliocene and early Pleistocene (e.g., Lawrence et al., 2006; Groeneveld et al., 2014). A comparison between modern and reconstructed Plio-/Pleistocene SSTs at Site 849 has already been presented in Jakob et al. (2017) and therefore will not be repeated in our manuscript.

While the depth habitat of *G. ruber* is relatively well understood (e.g., Wang, 2000) and therefore a comparison of *G. ruber*-based temperature estimates with modern values is possible, a comparison of *G. crassaformis*-based temperatures with modern values is not straight forward since the calcification depth of this species in the EEP has yet remained unclear. In the (sub-)tropical Atlantic, *G. crassaformis* typically calcifies between 400 and 800 m water depth (Steph et al., 2006; Regenberg et al., 2009; Steph et al., 2009, and references therein; Cléroux et al., 2013). Assuming the same calcification depth range for this species in the EEP, this corresponds to a modern temperature range of about ∼5–8 °C (Fig. 1b). *Globorotalia crassaformis*-based temperatures reconstructed for our study interval are similar to or somewhat higher than these values until MIS 100 (∼5–15 °C); thereafter, thermocline temperatures became somewhat lower (∼0–10 °C) (Fig. 5d). This would indicate that thermocline temperatures of the late Pliocene (early Pleistocene) were slightly higher (lower) than modern values, reflecting an overall warmer-than-present (colder-than-present)

EEP at thermocline depth. We are fully aware that this comparison is not straight forward in light of the uncertainties in the calcification depth of *G. crassaformis*. Therefore we decided not to include this comparison (i.e., modern *versus* late Pliocene/early Pleistocene thermocline temperatures) into our manuscript.

**R.2.16:** "note [table1 page 17]: Add the number of samples processed for each site and study to give a sense of the effort included in this study."
We appreciate this suggestion. We will include the number of samples processed (dried, weighed, washed) and geochemically analysed ($\delta^{13}C$, $\delta^{18}O$, Mg/Ca) per study. Note that the number of samples analysed is somewhat lower than the number of samples processed depending on the availability of foraminiferal (*G. crassaformis* and *G. ruber*) material.

**R.2.17:** "note [Figure 1 page 18, panel B]: A latitudinal transect would be more useful to test whether the front did change as in Koutavas et al."
We agree with Reviewer #2 that a latitudinal transect would be more useful to test changes in the frontal position. Given the amount of data that would be needed for such an approach, however, this would be clearly beyond the scope of this contribution (see also our response to comment R.2.4).

**References (other than those already cited in the discussion paper)**

Andreasen, D. J., and Ravelo, A. C.: Tropical Pacific Ocean thermocline depth reconstructions for the last glacial maximum, Paleocenaography, 3, 395–413, 1997.

Bahr, A., Nürnberg, D., Schönfeld, J., and Garbe-Schönberg, D.: Hydrological variability in Florida Straits during Marine Isotope Stage 5 cold events, Paleoceanography, 26, PA2214, doi:10.1029/2010PA002015, 2011.

Billups, K., Channell, J. E. T., and Zachos, J.: Late Oligocene to early Miocene geochronology and paleoceanography from the subantarctic South Atlantic. Paleoceanography, 17(1), 1004, doi:10.1029/2000PA000568, 2002.

Cléroux, C., and Lynch-Stieglitz, J.: What caused G. truncatulinoides to calcify in shallower water during the early Holocene in the western Atlantic/Gulf of Mexico?, IOP Conf. Ser. Earth Environ. Sci., 9, doi:10.1088/1755-1315/9/1/012020, 2010.

Craig, H., and Gordon, L. I.: Deuterium and oxygen-18 variations in the ocean and the marine atmosphere, in: Stable isotopes in oceanographic studies and paleotemperatures, edited by: Tongiorgi, E., Spoletto, Pisa, 9–130, 1965.

Davis, C. V., Badger, M. P. S., Bown, P. R., and Schmidt, D. N.: The response of calcifying plankton to climate change in the Pliocene, Biogeosciences, 10, 6131–6139, 2013.

DiNezio, P. N., Clement, A., Vecchi, G. A., Soden, B., Broccoli, A. J., Otto-Bliesner, B. L., and Braconnot, P.: The response of the Walker circulation to Last Glacial Maximum forcing: Implications for detection in proxies, Paleoceanography, 26, PA3217, doi:10.1029/ 2010PA002083, 2011.

Dittert, N., Baumann, K.-H., Bickert, T., Henrich, R., Huber, R., Kinkel, H., and Meggers, H.: Carbonate dissolution in the deep-sea: Methods, quantification and paleoceanographic application, in: Use of Proxies in Paleoceanography, edited by: Fischer, G. and Wefer, G., Springer, New York, 255–284,   1999.

Feldmeijer, W., Metcalfe, B., Brummer, G.-J. A., and Ganssen, G. M.: Reconstructing the depth of the permanent thermocline through the morphology and geochemistry of the deep dwelling planktonic foraminifer *Globorotalia truncatulinoides*, Paleoceanography and Paleoclimatology, 30, doi:10.1002/2014PA002687, 2015.

Ford, H. L., Ravelo, A. C., Polissar, P. J.: Reduced El Niño–Southern Oscillation during the Last Glacial Maximum, Science, 347, 255–258, 2015.

Hertzberg, J. E., Schmidt, M. W., Bianchi, T. S., Smith, R. W., Shields, M. R., and Marcantonio, F.: Comparison of eastern tropical Pacific $TEX_{86}$ and *Globigerinoides ruber* Mg/Ca derived sea surface temperatures: Insights from the Holocene and Last Glacial Maximum, Earth Planet. Sci. Lett, 434, 320–332, 2016.

Jehle, S., Bornemann, A., Deprez, A., and Speijer, R. P.: The impact of the Latest Danian Event on planktic foraminiferal faunas at ODP Site 1210 (Shatsky Rise, Pacific Ocean), PLoS One, doi:10.1371/journal.pone.0141644, 2015.

Karas, C., Nürnberg, D., Gupta, A. K., Tiedemann, R., Mohan, K., and Bickert, T.: Mid-Pliocene climate change amplified by a switch in Indonesian subsurface throughflow. Nat. Geosci., 2, 434–438, 2009.

Karas, C., Nürnberg, D., Tiedemann, R., and Garbe-Schönberg, D.: Pliocene climate change of the Southwest Pacific and the impact of ocean gateways, Earth Planet. Sci. Lett, 301, 117–124, 2011.

Luciani, V., D'Onofrio, R. D., Dickens, G. R., and Wade, B. S., Planktic foraminiferal response to early Eocene carbon cycle perturbations in the southeast Atlantic Ocean (ODP Site 1263), Global Planet. Change, 158, 119– 33, 2017.

Lynch-Stieglitz, J., Polissar, P. J., Jacobel, A. W., Hovan, S. A., Pockalny, R. A., Lyle, M., Murray, R. W., Ravelo, A. C., Bova, S. C., Dunlea, A. G., Ford, H. L., Hertzberg, J. E., Wertman, C. A., Maloney, A. E., Shackford, J. K., Wejnert, K., and Xie, R. C.: Glacial-interglacial changes in central tropical Pacific surface seawater property gradients, Paleoceanography, 30, doi:10.1002/2014PA002746, 2015.

Martínez-Garcia, A., Rosell-Melé, A-. McClymont, E. L., Gersonde, R., and Haug, G. H.: Subpolar link to the emergence of the modern equatorial Pacific cold tongue, Science, 328, 5985, 1550–1553, 2010.

Pusz, A. E., Thunell, R. C., and Miller, K. G. (2011). Deep water temperature, carbonate ion, and ice volume changes across the Eocene-Oligocene climate transition. Paleoceanography, 26, PA2205, doi:10.1029/2010PA001950, 2011.

Sexton, P. F., and Norris, R. D.: Dispersal and biogeography of marine plankton: Long-distance dispersal of the foraminifer *Truncorotalia truncatulinoides*, Geology, 36, 899–902, 2008.